# Information content differentiates enhancers from silencers in mouse photoreceptors

Ryan Z Friedman[1,2], David M Granas[1,2], Connie A Myers[3], Joseph C Corbo[3], Barak A Cohen[1,2], Michael A White[1,2]*

[1]Edison Family Center for Genome Sciences and Systems Biology, Washington University School of Medicine, St. Louis, United States; [2]Department of Genetics, Washington University School of Medicine, St. Louis, United States; [3]Department of Pathology and Immunology, Washington University School of Medicine, St Louis, United States

**Abstract** Enhancers and silencers often depend on the same transcription factors (TFs) and are conflated in genomic assays of TF binding or chromatin state. To identify sequence features that distinguish enhancers and silencers, we assayed massively parallel reporter libraries of genomic sequences targeted by the photoreceptor TF cone-rod homeobox (CRX) in mouse retinas. Both enhancers and silencers contain more TF motifs than inactive sequences, but relative to silencers, enhancers contain motifs from a more diverse collection of TFs. We developed a measure of information content that describes the number and diversity of motifs in a sequence and found that, while both enhancers and silencers depend on CRX motifs, enhancers have higher information content. The ability of information content to distinguish enhancers and silencers targeted by the same TF illustrates how motif context determines the activity of *cis*-regulatory sequences.

*For correspondence:
mawhite@wustl.edu

**Competing interest:** The authors declare that no competing interests exist.

## Introduction

Active *cis*-regulatory sequences in the genome are characterized by accessible chromatin and specific histone modifications, which reflect the action of DNA-binding transcription factors (TFs) that recognize specific sequence motifs and recruit chromatin-modifying enzymes (*Klemm et al., 2019*). These epigenetic hallmarks of active chromatin are routinely used to train machine learning models that predict *cis*-regulatory sequences, based on the assumption that such epigenetic marks are reliable predictors of genuine *cis*-regulatory sequences (*Ernst and Kellis, 2012*; *Ghandi et al., 2014*; *Hoffman et al., 2012*; *Kelley et al., 2016*; *Lee et al., 2011*; *Sethi et al., 2020*; *Zhou and Troyanskaya, 2015*). However, results from functional assays show that many predicted *cis*-regulatory sequences exhibit little or no *cis*-regulatory activity. Typically, 50 % or more of predicted *cis*-regulatory sequences fail to drive expression in massively parallel reporter assays (MPRAs) (*Moore et al., 2020*; *Kwasnieski et al., 2014*), indicating that an active chromatin state is not sufficient to reliably identify *cis*-regulatory sequences.

Another challenge is that enhancers and silencers are difficult to distinguish by chromatin accessibility or epigenetic state (*Doni Jayavelu et al., 2020*; *Gisselbrecht et al., 2020*; *Pang and Snyder, 2020*; *Petrykowska et al., 2008*; *Segert et al., 2021*), and thus computational predictions of *cis*-regulatory sequences often do not differentiate between enhancers and silencers. Silencers are often enhancers in other cell types (*Brand et al., 1987*; *Doni Jayavelu et al., 2020*; *Gisselbrecht et al., 2020*; *Huang et al., 2021*; *Jiang et al., 1993*; *Ngan et al., 2020*; *Pang and Snyder, 2020*), reside in open chromatin (*Doni Jayavelu et al., 2020*; *Huang et al., 2019*; *Huang et al., 2021*; *Pang and*

**eLife digest** Different cell types are established by activating and repressing the activity of specific sets of genes, a process controlled by proteins called transcription factors. Transcription factors work by recognizing and binding short stretches of DNA in parts of the genome called cis-regulatory sequences. A cis-regulatory sequence that increases the activity of a gene when bound by transcription factors is called an enhancer, while a sequence that causes a decrease in gene activity is called a silencer.

To establish a cell type, a particular transcription factor will act on both enhancers and silencers that control the activity of different genes. For example, the transcription factor cone-rod homeobox (CRX) is critical for specifying different types of cells in the retina, and it acts on both enhancers and silencers. In rod photoreceptors, CRX activates rod genes by binding their enhancers, while repressing cone photoreceptor genes by binding their silencers. However, CRX always recognizes and binds to the same DNA sequence, known as its binding site, making it unclear why some cis-regulatory sequences bound to CRX act as silencers, while others act as enhancers.

Friedman et al. sought to understand how enhancers and silencers, both bound by CRX, can have different effects on the genes they control. Since both enhancers and silencers contain CRX binding sites, the difference between the two must lie in the sequence of the DNA surrounding these binding sites.

Using retinas that have been explanted from mice and kept alive in the laboratory, Friedman et al. tested the activity of thousands of CRX-binding sequences from the mouse genome. This showed that both enhancers and silencers have more copies of CRX-binding sites than sequences of the genome that are inactive. Additionally, the results revealed that enhancers have a diverse collection of binding sites for other transcription factors, while silencers do not. Friedman et al. developed a new metric they called information content, which captures the diverse combinations of different transcription binding sites that cis-regulatory sequences can have. Using this metric, Friedman et al. showed that it is possible to distinguish enhancers from silencers based on their information content.

It is critical to understand how the DNA sequences of cis-regulatory regions determine their activity, because mutations in these regions of the genome can cause disease. However, since every person has thousands of benign mutations in cis-regulatory sequences, it is a challenge to identify specific disease-causing mutations, which are relatively rare. One long-term goal of models of enhancers and silencers, such as Friedman et al.'s information content model, is to understand how mutations can affect cis-regulatory sequences, and, in some cases, lead to disease.

---

*Snyder, 2020*), sometimes bear epigenetic marks of active enhancers (*Fan et al., 2016*; *Huang et al., 2021*), and can be bound by TFs that also act on enhancers in the same cell type (*Alexandre and Vincent, 2003*; *Grass et al., 2003*; *Huang et al., 2021*; *Iype et al., 2004*; *Jiang et al., 1993*; *Liu et al., 2014*; *Martínez-Montañés et al., 2013*; *Peng et al., 2005*; *Rachmin et al., 2015*; *Rister et al., 2015*; *Stampfel et al., 2015*; *White et al., 2013*). As a result, enhancers and silencers share similar sequence features, and understanding how they are distinguished in a particular cell type remains an important challenge (*Segert et al., 2021*).

The TF cone-rod homeobox (CRX) controls selective gene expression in a number of different photoreceptor and bipolar cell types in the retina (*Chen et al., 1997*; *Freund et al., 1997*; *Furukawa et al., 1997*; *Murphy et al., 2019*). These cell types derive from the same progenitor cell population (*Koike et al., 2007*; *Wang et al., 2014*), but they exhibit divergent, CRX-directed transcriptional programs (*Corbo et al., 2010*; *Hennig et al., 2008*; *Hughes et al., 2017*; *Murphy et al., 2019*). CRX cooperates with cell type-specific co-factors to selectively activate and repress different genes in different cell types and is required for differentiation of rod and cone photoreceptors (*Chen et al., 2005*; *Hao et al., 2012*; *Hennig et al., 2008*; *Hsiau et al., 2007*; *Irie et al., 2015*; *Kimura et al., 2000*; *Lerner et al., 2005*; *Mears et al., 2001*; *Mitton et al., 2000*; *Murphy et al., 2019*; *Peng et al., 2005*; *Sanuki et al., 2010*; *Srinivas et al., 2006*). However, the sequence features that define CRX-targeted enhancers vs. silencers in the retina are largely unknown.

We previously found that a significant minority of CRX-bound sequences act as silencers in an MPRA conducted in live mouse retinas (*White et al., 2013*), and that silencer activity requires CRX

(*White et al., 2016*). Here, we extend our analysis by testing thousands of additional candidate *cis*-regulatory sequences. We show that while regions of accessible chromatin and CRX binding exhibit a range of *cis*-regulatory activity, enhancers and silencers contain more TF motifs than inactive sequences, and that enhancers are distinguished from silencers by a higher diversity of TF motifs. We capture the differences between these sequence classes with a new metric, motif information content (Boltzmann entropy), that considers only the number and diversity of TF motifs in a candidate *cis*-regulatory sequence. Our results suggest that CRX-targeted enhancers are defined by a flexible regulatory grammar and demonstrate how differences in motif information content encode functional differences between genomic loci with similar chromatin states.

## Results

We tested the activities of 4844 putative CRX-targeted *cis*-regulatory sequences (CRX-targeted sequences) by MPRA in live retinas. The MPRA libraries consist of 164 bp genomic sequences centered on the best match to the CRX position weight matrix (PWM) (*Lee et al., 2010*) whenever a CRX motif is present, and matched sequences in which all CRX motifs were abolished by point mutation (Materials and methods). The MPRA libraries include 3299 CRX-bound sequences identified by ChIP-seq in the adult retina (*Corbo et al., 2010*) and 1545 sequences that do not have measurable CRX binding in the adult retina but reside in accessible chromatin in adult photoreceptors (*Hughes et al., 2017*) and have the H3K27ac enhancer mark in postnatal day 14 (P14) retina (*Ruzycki et al., 2018*) ('ATAC-seq peaks'). We split the sequences across two plasmid libraries, each of which contained the same 150 scrambled sequences as internal controls (*Supplementary files 1 and 2*). We cloned sequences upstream of the rod photoreceptor-specific *Rhodopsin* (*Rho*) promoter and a *DsRed* reporter gene, electroporated libraries into explanted mouse retinas at P0 in triplicate, harvested the retinas at P8, and then sequenced the RNA and input DNA plasmid pool. The data is highly reproducible across replicates ($R^2 > 0.96$, *Figure 1—figure supplement 1*). After activity scores were calculated and normalized to the basal *Rho* promoter, the two libraries were well calibrated and merged together (two-sample Kolmogorov-Smirnov test p = 0.09, *Figure 1—figure supplement 2*, *Supplementary file 3*, and Materials and methods).

### Strong enhancers and silencers have high CRX motif content

The *cis*-regulatory activities of CRX-targeted sequences vary widely (*Figure 1a*). We defined enhancers and silencers as those sequences that have statistically significant activity that is at least twofold above or below the activity of the basal *Rho* promoter (Welch's t-test, Benjamini-Hochberg false discovery rate (FDR) q < 0.05, *Supplementary file 3*). We defined inactive sequences as those whose activity is both within a twofold change of basal activity and not significantly different from the basal *Rho* promoter. We further stratified enhancers into strong and weak enhancers based on whether or not they fell above the 95th percentile of scrambled sequences. Using these criteria, 22 % of CRX-targeted sequences are strong enhancers, 28 % are weak enhancers, 19 % are inactive, and 17 % are silencers; the remaining 13 % were considered ambiguous and removed from further analysis. To test whether these sequences function as CRX-dependent enhancers and silencers in the genome, we examined genes differentially expressed in *Crx*[-/-] retina (*Roger et al., 2014*). Genes that are de-repressed are more likely to be near silencers (Fisher's exact test p = 0.001, odds ratio = 2.1, n = 206) and genes that are down-regulated are more likely to be near enhancers (Fisher's exact test p = 0.02, odds ratio = 1.5, n = 344, Materials and methods), suggesting that our reporter assay identified sequences that act as enhancers and silencers in the genome. We sought to identify features that would accurately classify these different classes of sequences.

Neither CRX ChIP-seq-binding status nor DNA accessibility as measured by ATAC-seq strongly differentiates between these four classes (*Figure 1b*). Compared to CRX ChIP-seq peaks, ATAC-seq peaks that lack CRX binding in the adult retina are slightly enriched for inactive sequences (Fisher's exact test p = $2 \times 10^{-7}$, odds ratio = 1.5) and slightly depleted for strong enhancers (Fisher's exact test p = $1 \times 10^{-21}$, odds ratio = 2.2). However, sequences with ChIP-seq or ATAC-seq peaks span all four activity categories, consistent with prior reports that DNA accessibility and TF binding data are not sufficient to identify functional enhancers and silencers (*Doni Jayavelu et al., 2020*; *Huang et al., 2019*; *Huang et al., 2021*; *Pang and Snyder, 2020*; *White et al., 2013*).

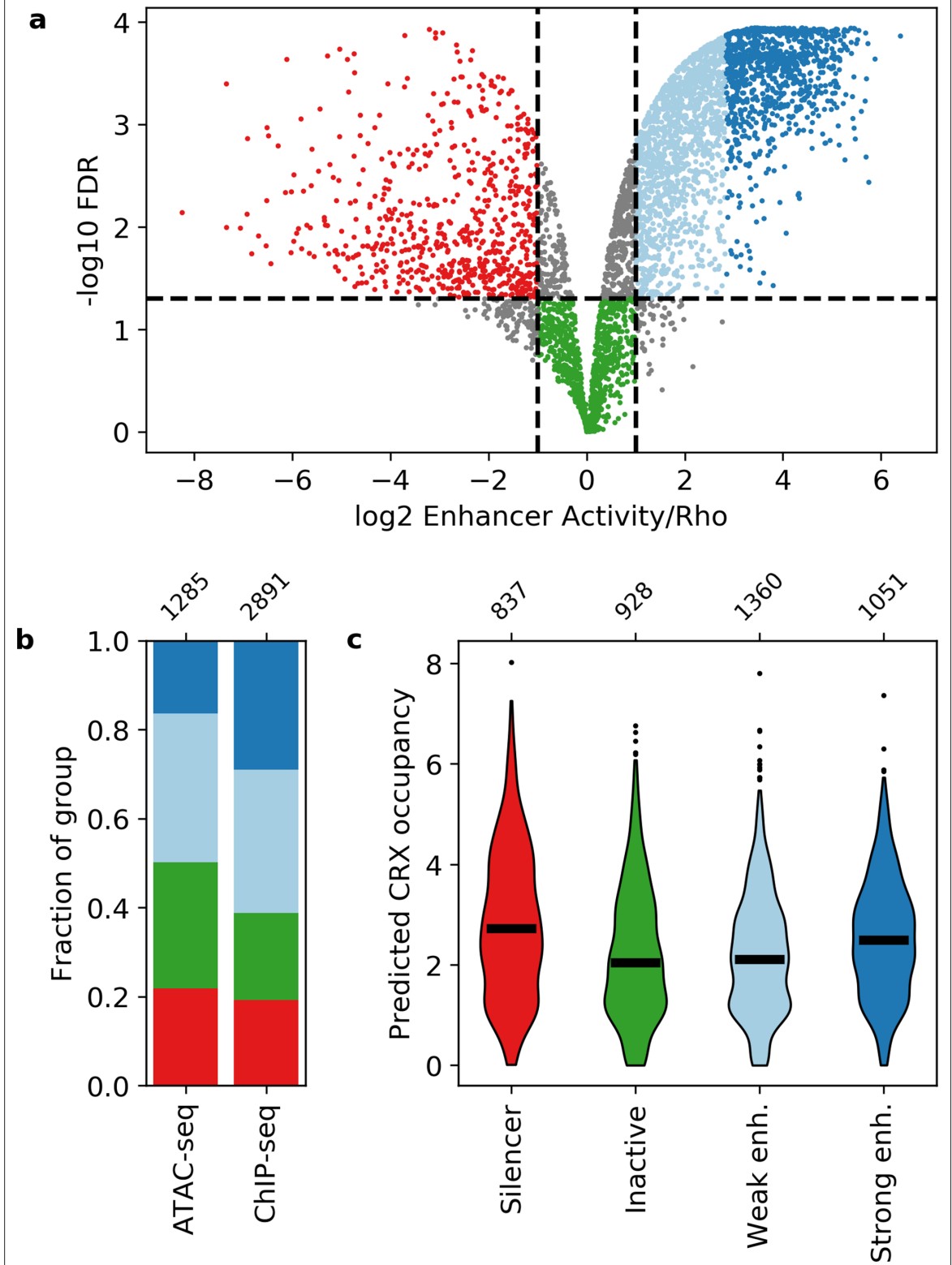

**Figure 1.** Activity of putative *cis*-regulatory sequences with cone-rod homeobox (CRX) motifs. (**a**) Volcano plot of activity scores relative to the *Rho* promoter alone. Sequences are grouped as strong enhancers (dark blue), weak enhancers (light blue), inactive (green), silencers (red), or ambiguous (gray). Horizontal line, false discovery rate (FDR) q = 0.05. Vertical lines, twofold above and below *Rho*. (**b**) Fraction of ChIP-seq and ATAC-seq peaks that belong to each activity group. (**c**) Predicted CRX occupancy of each activity group. Horizontal lines, medians; enh., enhancer. Numbers at top of (**b and c**) indicate n for groups.

*Figure 1 continued on next page*

*Figure 1 continued*

The online version of this article includes the following figure supplement(s) for figure 1:

**Figure supplement 1.** Reproducibility of massively parallel reporter assay (MPRA) measurements.

**Figure supplement 2.** Calibration of massively parallel reporter assay (MPRA) libraries with the *Rho* promoter.

We examined whether the number and affinity of CRX motifs differentiate enhancers, silencers, and inactive sequences by computing the predicted CRX occupancy (i.e. expected number of bound molecules) for each sequence (**White et al., 2013**). Consistent with our previous work (**White et al., 2016**), both strong enhancers and silencers have higher predicted CRX occupancy than inactive sequences (Mann-Whitney U test, $p = 6 \times 10^{-10}$ and $6 \times 10^{-17}$, respectively, **Figure 1c**), suggesting that total CRX motif content helps distinguish silencers and strong enhancers from inactive sequences. However, predicted CRX occupancy does not distinguish strong enhancers from silencers: a logistic regression classifier trained with fivefold cross-validation only achieves an area under the receiver operating characteristic (AUROC) curve of 0.548 ± 0.023 and an area under the precision recall (AUPR) curve of 0.571 ± 0.020 (**Figure 2a** and **Figure 2—figure supplement 1**). We thus sought to identify sequence features that distinguish strong enhancers from silencers.

## Lineage-defining TF motifs differentiate strong enhancers from silencers

We performed a de novo motif enrichment analysis to identify motifs that distinguish strong enhancers from silencers and found several differentially enriched motifs matching known TFs. For motifs that matched multiple TFs, we selected one representative TF for downstream analysis, since TFs from the same family have PWMs that are too similar to meaningfully distinguish between motifs for these TFs (**Figure 2—figure supplement 2**, Materials and methods). Strong enhancers are enriched for several motif families that include TFs that interact with CRX or are important for photoreceptor development: NeuroD1/NDF1 (E-box-binding bHLH) (**Morrow et al., 1999**), RORB (nuclear receptor) (**Jia et al., 2009**; **Srinivas et al., 2006**), MAZ or Sp4 (C2H2 zinc finger) (**Lerner et al., 2005**), and NRL (bZIP) (**Mears et al., 2001**; **Mitton et al., 2000**). Sp4 physically interacts with CRX in the retina (**Lerner et al., 2005**), but we chose to represent the zinc finger motif with MAZ because it has a higher quality score in the HOCOMOCO database (**Kulakovskiy et al., 2018**). Silencers were enriched for a motif that resembles a partial K50 homeodomain motif but instead matches the zinc finger TF GFI1, a member of the Snail repressor family (**Chiang and Ayyanathan, 2013**) expressed in developing retinal ganglion cells (**Yang et al., 2003**). Therefore, while strong enhancers and silencers are not distinguished by their CRX motif content, strong enhancers are uniquely enriched for several lineage-defining TFs.

To quantify how well these TF motifs differentiate strong enhancers from silencers, we trained two different classification models with fivefold cross-validation. First, we trained a 6-mer support vector machine (SVM) (**Ghandi et al., 2014**) and achieved an AUROC of 0.781 ± 0.013 and AUPR of 0.812 ± 0.020 (**Figure 2a** and **Figure 2—figure supplement 1**). The SVM considers all 2080 non-redundant 6-mers and provides an upper bound to the predictive power of models that do not consider the exact arrangement or spacing of sequence features. We next trained a logistic regression model on the predicted occupancy for eight lineage-defining TFs (**Supplementary file 4**) and compared it to the upper bound established by the SVM. In this model, we considered CRX, the five TFs identified in our motif enrichment analysis, and two additional TFs enriched in photoreceptor ATAC-seq peaks (**Hughes et al., 2017**): RAX, a Q50 homeodomain TF that contrasts with CRX, a K50 homeodomain TF (**Irie et al., 2015**) and MEF2D, a MADS box TF which co-binds with CRX (**Andzelm et al., 2015**). The logistic regression model performs nearly as well as the SVM (AUROC 0.698 ± 0.036, AUPR 0.745 ± 0.032, **Figure 2a** and **Figure 2—figure supplement 1**) despite a 260-fold reduction from 2080 to 8 features. To determine whether the logistic regression model depends specifically on the eight lineage-defining TFs, we established a null distribution by fitting 100 logistic regression models with randomly selected TFs (Materials and methods). Our logistic regression model outperforms the null distribution (one-tailed Z-test for AUROC and AUPR, $p < 0.0008$, **Figure 2—figure supplement 3**), indicating that the performance of the model specifically requires the eight lineage-defining TFs. To determine whether the SVM identified any additional motifs that could be added to the logistic

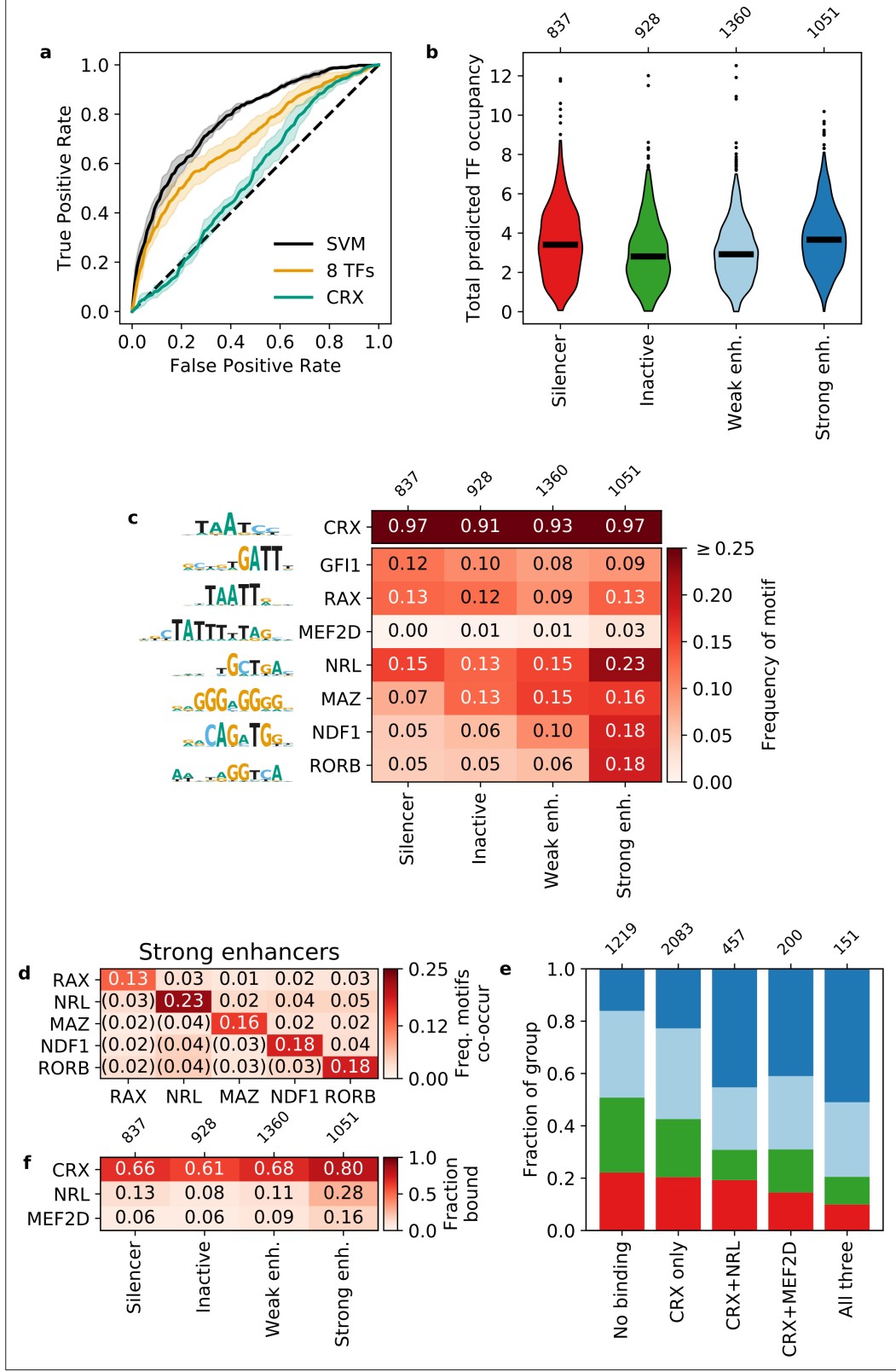

**Figure 2.** Strong enhancers contain a diverse array of motifs. (**a**) Receiver operating characteristic for classifying strong enhancers from silencers. Solid black, 6-mer support vector machine (SVM); orange, eight transcription factors (TFs) predicted occupancy logistic regression; aqua, predicted cone-rod homeobox (CRX) occupancy logistic regression; dashed black, chance; shaded area, 1 standard deviation based on fivefold cross-validation.

*Figure 2 continued on next page*

*Figure 2 continued*

(**b and c**) Total predicted TF occupancy (**b**) and frequency of TF motifs (**c**) in each activity class. (**d**) Frequency of co-occurring TF motifs in strong enhancers. Lower triangle is expected co-occurrence if motifs are independent. (**e**) Frequency of activity classes, colored as in (**b**), for sequences in CRX, NRL, and/or MEF2D ChIP-seq peaks. (**f**) Frequency of TF ChIP-seq peaks in activity classes. TFs in (**c**) are sorted by feature importance of the logistic regression model in (**a**).

The online version of this article includes the following figure supplement(s) for figure 2:

**Figure supplement 1.** Precision recall curve for strong enhancer vs. silencer classifiers.

**Figure supplement 2.** Results from de novo motif analysis.

**Figure supplement 3.** Additional validation of the eight transcription factors (TFs) predicted occupancy logistic regression model.

---

regression model, we generated de novo motifs using the SVM *k*-mer scores and found no additional motifs predictive of strong enhancers. Finally, we found that our two models perform similarly on an independent test set of CRX-targeted sequences (*White et al., 2013*; *Figure 2—figure supplement 3*). Since the logistic regression model performs near the upper bound established by the SVM and depends specifically on the eight selected motifs, we conclude that these motifs comprise nearly all of the sequence features captured by the SVM that distinguish strong enhancers from silencers among CRX-targeted sequences.

## Strong enhancers are characterized by diverse total motif content

To understand how these eight TF motifs differentiate strong enhancers from silencers, we first calculated the total predicted occupancy of each sequence by all eight lineage-defining TFs and compared the different activity classes. Strong enhancers and silencers both have higher total predicted occupancies than inactive sequences, but total predicted occupancies do not distinguish strong enhancers and silencers from each other (*Figure 2b*, *Supplementary file 5*). Since strong enhancers are enriched for several motifs relative to silencers, this suggests that strong enhancers are distinguished from silencers by the diversity of their motifs, rather than the total number.

We considered two hypotheses for how the more diverse collection of motifs function in strong enhancers: either strong enhancers depend on specific combinations of TF motifs ('TF identity hypothesis') or they instead must be co-occupied by multiple lineage-defining TFs, regardless of TF identity ('TF diversity hypothesis'). To distinguish between these hypotheses, we examined which specific motifs contribute to the total motif content of strong enhancers and silencers. We considered motifs for a TF present in a sequence if the TF predicted occupancy was above 0.5 molecules (*Supplementary file 4*), which generally corresponds to at least one motif with a relative $K_D$ above 3 %. This threshold captures the effect of low affinity motifs that are often biologically relevant (*Crocker et al., 2015*; *Farley et al., 2015*; *Farley et al., 2016*; *Parker et al., 2011*). As expected, 97 % of strong enhancers and silencers contain CRX motifs since the sequences were selected based on CRX binding or significant matches to the CRX PWM within open chromatin (*Figure 2c*). Compared to silencers, strong enhancers contain a broader diversity of motifs for the eight lineage-defining TFs (*Figure 2c*). However, while strong enhancers contain a broader range of motifs, no single motif occurs in a majority of strong enhancers: NRL motifs are present in 23 % of strong enhancers, NeuroD1 and RORB in 18 % each, and MAZ in 16 %. Additionally, none of the motifs tend to co-occur as pairs in strong enhancers: no specific pair occurred in more than 5 % of sequences (*Figure 2d*). We also did not observe a bias in the linear arrangement of motifs in strong enhancers (Materials and methods). Similarly, no single motif occurs in more than 15 % of silencers (*Figure 2c*). These results suggest that strong enhancers are defined by the diversity of their motifs, and not by specific motif combinations or their linear arrangement.

The results above predict that strong enhancers are more likely to be bound by a diverse but degenerate collection of TFs, compared with silencers or inactive sequences. We tested this prediction by examining in vivo TF binding using published ChIP-seq data for NRL (*Hao et al., 2012*) and MEF2D (*Andzelm et al., 2015*). Consistent with the prediction, sequences bound by CRX and either NRL or MEF2D are approximately twice as likely to be strong enhancers compared to sequences only bound by CRX (*Figure 2e*). Sequences bound by all three TFs are the most likely to be strong or weak

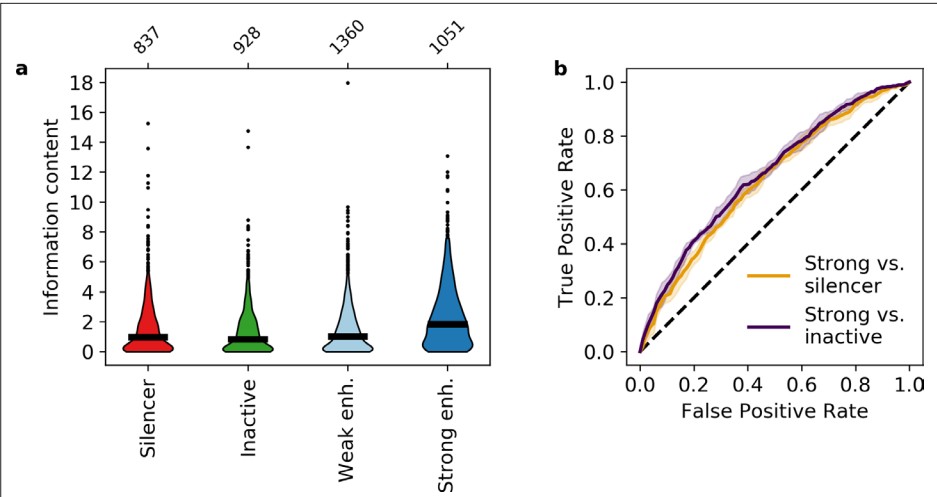

**Figure 3.** Information content classifies strong enhancers. (**a**) Information content for different activity classes. (**b**) Receiver operating characteristic of information content to classify strong enhancers from silencers (orange) or inactive sequences (indigo).

The online version of this article includes the following figure supplement(s) for figure 3:

**Figure supplement 1.** Precision recall curve of logistic regression classifier using information content.

enhancers rather than silencers or inactive sequences. However, most strong enhancers are not bound by either NRL or MEF2D (**Figure 2f**), indicating that binding of these TFs is not required for strong enhancers. Our results support the TF diversity hypothesis: CRX-targeted enhancers are co-occupied by multiple TFs, without a requirement for specific combinations of lineage-defining TFs.

## Strong enhancers have higher motif information content than silencers

Our results indicate that both strong enhancers and silencers have a higher total motif content than inactive sequences, while strong enhancers contain a more diverse collection of motifs than silencers. To quantify these differences in the number and diversity of motifs, we computed the information content of CRX-targeted sequences using Boltzmann entropy. The Boltzmann entropy of a system is related to the number of ways the system's molecules can be arranged, which increases with either the number or diversity of molecules (**Phillips et al., 2012**, Chapter 5). In our case, each TF is a different type of molecule and the number of each TF is represented by its predicted occupancy for a *cis*-regulatory sequence. The number of molecular arrangements is thus $W$, the number of distinguishable permutations that the TFs can be ordered on the sequence, and the information content of a sequence is then $\log_2 W$ (Materials and methods).

We found that on average, strong enhancers have higher information content than both silencers and inactive sequences (Mann-Whitney U test, p = $1 \times 10^{-23}$ and $7 \times 10^{-34}$, respectively, **Figure 3a**, **Supplementary file 5**), confirming that information content captures the effect of both the number and diversity of motifs. Quantitatively, the average silencer and inactive sequence contains 1.6 and 1.4 bits, respectively, which represents approximately three total motifs for two TFs. Strong enhancers contain on average 2.4 bits, representing approximately three total motifs for three TFs or four total motifs for two TFs. To compare the predictive value of our information content metric to the model based on all eight motifs, we trained a logistic regression model and found that information content classifies strong enhancers from silencers with an AUROC of 0.634 ± 0.008 and an AUPR of 0.663 ± 0.014 (**Figure 3b** and **Figure 3—figure supplement 1**). This is only slightly worse than the model trained on eight TF occupancies despite an eightfold reduction in the number of features, which is itself comparable to the SVM with 2080 features. The difference between the two logistic regression models suggests that the specific identities of TF motifs make some contribution to the eight TF model, but that most of the signal captured by the SVM can be described with a single metric that does not assign weights to specific motifs. Information content also distinguishes strong enhancers from inactive sequences (AUROC 0.658 ± 0.012, AUPR 0.675 ± 0.019, **Figure 3b** and **Figure 3—figure**

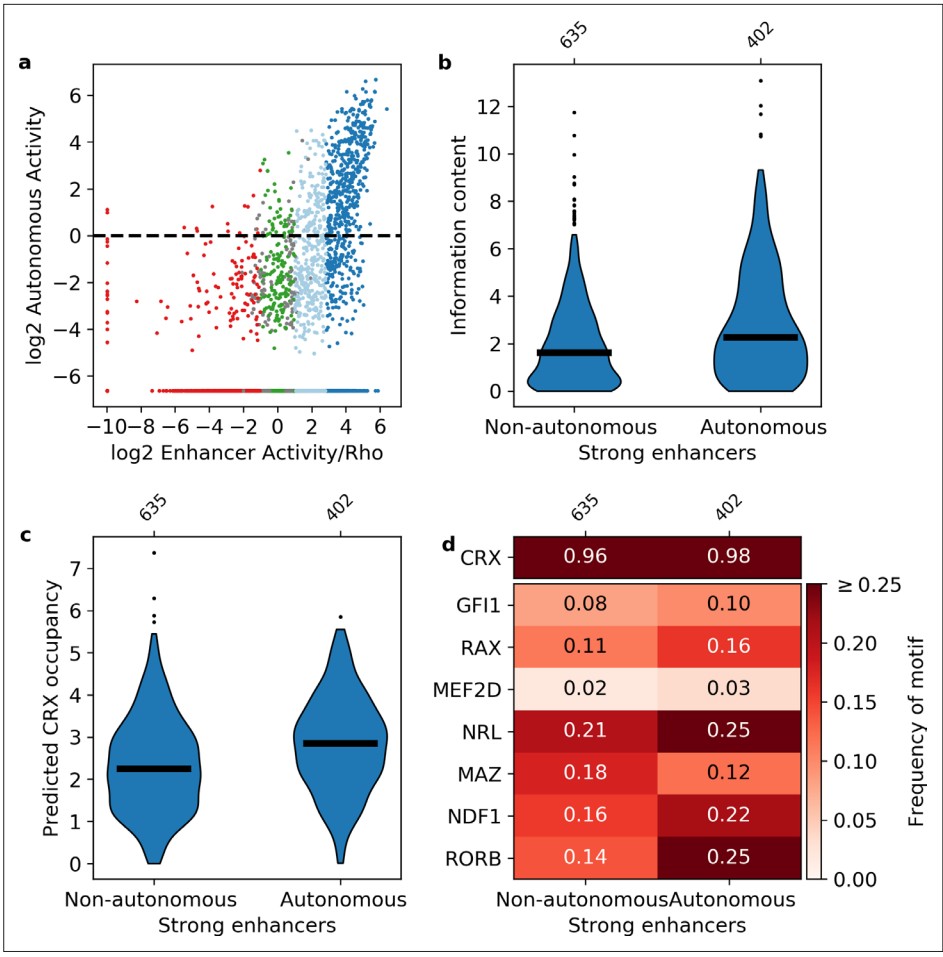

**Figure 4.** Sequence features of autonomous and non-autonomous strong enhancers. (**a**) Activity of library in the presence (x-axis) or absence (y-axis) of the *Rho* promoter. Dark blue, strong enhancers; light blue, weak enhancers; green, inactive; red, silencers; gray, ambiguous; horizontal line, cutoff for autonomous activity. Points on the far left and/or very bottom are sequences that were present in the plasmid pool but not detected in the RNA. (**b–d**) Comparison of autonomous and non-autonomous strong enhancers for information content (**b**), predicted cone-rod homeobox (CRX) occupancy (**c**), and frequency of transcription factor (TF) motifs (**d**).

supplement 1). These results indicate that strong enhancers are characterized by higher information content, which reflects both the total number and diversity of motifs.

## Strong enhancers require high information content but not NRL motifs

Our results show that except for CRX, none of the lineage-defining motifs occur in a majority of strong enhancers. However, all sequences were tested in reporter constructs with the *Rho* promoter, which contains an NRL motif and three CRX motifs (*Corbo et al., 2010*; *Kwasnieski et al., 2012*). Since NRL is a key co-regulator with CRX in rod photoreceptors, we tested whether strong enhancers generally require NRL, which would be inconsistent with our TF diversity hypothesis. We removed the NRL motif by recloning our MPRA library without the basal *Rho* promoter. If strong enhancers require an NRL motif for high activity, then only CRX-targeted sequences with NRL motifs will drive reporter expression. If information content (i.e. total motif content and diversity) is the primary determinant of strong enhancers, only CRX-targeted sequences with sufficient motif diversity, measured by information content, will drive reporter expression regardless of whether or not NRL motifs are present.

We replaced the *Rho* promoter with a minimal 23 bp polylinker sequence between our libraries and *DsRed*, and repeated the MPRA (*Figure 1—figure supplement 1*, *Supplementary file 3*). CRX-targeted sequences were designated as 'autonomous' if they retained activity in the absence of the *Rho* promoter (log₂(RNA/DNA) > 0, Materials and methods). We found that 90 % of autonomous

sequences are from the enhancer class, while less than 3 % of autonomous sequences are from the silencer class (*Figure 4a*). This confirms that the distinction between silencers and enhancers does not depend on the *Rho* promoter, which is consistent with our previous finding that CRX-targeted silencers repress other promoters (*Hughes et al., 2018*; *White et al., 2016*). However, while most autonomous sequences are enhancers, only 39 % of strong enhancers and 9 % of weak enhancers act autonomously. Consistent with a role for information content, autonomous strong enhancers have higher information content (Mann-Whitney U test p = 4 × 10⁻⁸, *Figure 4b*) and higher predicted CRX occupancy (Mann-Whitney U test p = 9 × 10⁻¹², *Figure 4c*) than non-autonomous strong enhancers. We found no evidence that specific lineage-defining motifs are required for autonomous activity, including NRL, which is present in only 25 % of autonomous strong enhancers (*Figure 4d*). Similarly, NRL ChIP-seq binding (*Hao et al., 2012*) occurs more often among autonomous strong enhancers (41% vs. 19%, Fisher's exact test p = 2 × 10⁻¹⁴, odds ratio = 3.0), yet NRL binding still only accounts for a minority of these sequences. We thus conclude that strong enhancers require high information content, rather than any specific lineage-defining motifs.

## TF motifs contribute independently to strong enhancers

Our results indicate that information content distinguishes strong enhancers from silencers and inactive sequences. Information content only takes into account the total number and diversity of motifs in a sequence and not any potential interactions between them. The classification success of information content thus suggests that each TF motif will contribute independently to enhancer activity. We tested this prediction with CRX-targeted sequences where all CRX motifs were abolished by point mutation (*Supplementary file 3*). Consistent with our previous work (*White et al., 2013*), mutating CRX motifs causes the activities of both enhancers and silencers to regress toward basal levels (Pearson's r = 0.608, *Figure 5a*), indicating that most enhancers and silencers show some dependence on CRX. However, 40 % of wild-type strong enhancers show low CRX dependence and remain strong enhancers with their CRX motifs abolished. Although strong enhancers with high and low CRX dependence have similar wild-type information content (*Figure 5b*), strong enhancers with low CRX dependence have lower predicted CRX occupancy than those with high CRX dependence (Mann-Whitney U test p = 2 × 10⁻⁹, *Figure 5c*), and also have higher 'residual' information content (i.e. information content without CRX motifs, Mann-Whitney U test p = 1 × 10⁻⁷, *Figure 5d*). Low CRX dependence sequences have an average of 1.5 residual bits, which corresponds to three motifs for two TFs, while high CRX dependence sequences have an average of 1.0 residual bits, which corresponds to two motifs for two TFs (*Figure 5e*).

Strong enhancers with low and high CRX dependence have similar wild-type information content and similar total predicted occupancy (*Figure 5b and e*). As a result, sequences with more CRX motifs have fewer motifs for other TFs, suggesting that there is no evolutionary pressure for enhancers to contain additional motifs beyond the minimum amount of information content required to be active. To test this idea, we calculated the minimum number and diversity of motifs necessary to specify a relatively unique location in the genome (*Wunderlich and Mirny, 2009*) and found that a 164 bp sequence only requires five motifs for three TFs (Materials and methods). These motif requirements can be achieved in two ways with similar information content that differ only in the quantitative number of motifs for each TF. In other words, the number of motifs for any particular TF is not important so long as there is sufficient information content. Taken together, we conclude that each TF motif provides an independent contribution toward specifying strong enhancers.

## Discussion

Many regions in the genome are bound by TFs and bear the epigenetic hallmarks of active *cis*-regulatory sequences, yet fail to exhibit *cis*-regulatory activity when tested directly. The discrepancy between measured epigenomic state and *cis*-regulatory activity indicates that enhancers and silencers consist of more than the minimal sequence features necessary to recruit TFs and chromatin-modifying factors. Our results show that enhancers, silencers, and inactive sequences in developing photoreceptors can be distinguished by their motif content, even though they are indistinguishable by CRX binding or chromatin accessibility. We show that both enhancers and silencers contain more TF motifs than inactive sequences, and that enhancers also contain more diverse sets of motifs for lineage-defining

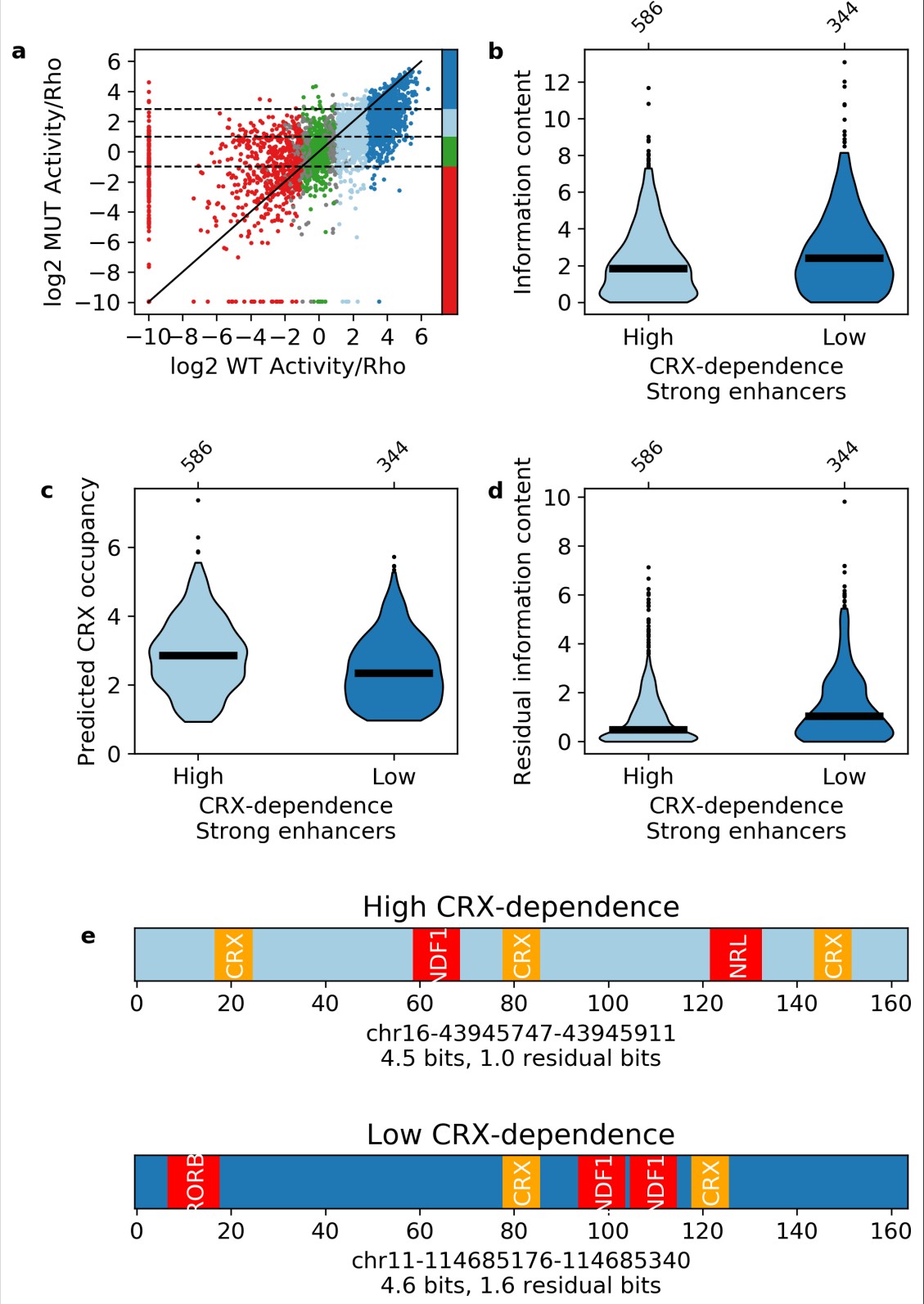

**Figure 5.** Independence of transcription factor (TF) motifs in strong enhancers. (**a**) Activity of sequences with and without cone-rod homeobox (CRX) motifs. Points are colored by the activity group with CRX motifs intact: dark blue, strong enhancers; light blue, weak enhancers; green, inactive; red, silencers; gray, ambiguous; horizontal dotted lines and color bar represent the cutoffs for the same groups when CRX motifs are mutated. Solid black line is the y = x line. (**b–d**) Comparison of strong enhancers with high and low CRX dependence for information content (**b**), predicted CRX occupancy (**c**), and residual information content (**d**). (**e**) Representative strong enhancers with high (top) or low (bottom) CRX dependence.

TFs. These differences are captured by our measure of information content. Information content, as a single metric, identifies strong enhancers nearly as well as an unbiased set of 2080 non-redundant 6-mers used for an SVM, indicating that a simple measure of motif number and diversity can capture the key sequence features that distinguish enhancers from other sequences that lie in open chromatin.

The results of our information content classifier are consistent with the TF collective model of enhancers (*Junion et al., 2012*; *Spitz and Furlong, 2012*): globally, active enhancers are specified by the combinatorial action of lineage-defining TFs with little constraint on which motifs must co-occur. We show that CRX-targeted enhancers are distinguished from inactive CRX-targeted sequences by a larger, more diverse collection of TF motifs, and not any specific combination of motifs. This indicates that enhancers are active because they have acquired the necessary number of TF binding motifs, and not because they are defined by a strict regulatory grammar. Sequences with fewer motifs may be bound by CRX and reside within open chromatin, but they lack sufficient TF binding for activity. Such loose constraints would facilitate the de novo emergence of tissue-specific enhancers and silencers over evolution and explain why critical cell type-specific TF interactions, such as CRX and NRL in rod photoreceptors, occur at only a minority of the active enhancers in that cell type (*Hsiau et al., 2007*; *Hughes et al., 2018*; *White et al., 2013*).

Like enhancers, CRX-targeted silencers require higher motif content and are dependent on CRX motifs, but they lack the TF diversity of enhancers. The lack of TF diversity in silencers parallels the architecture of signal-responsive *cis*-regulatory sequences, which are silencers in the absence of a signal and require multiple activators for induction (*Barolo and Posakony, 2002*). Consistent with this, we previously showed using synthetic sequences that high occupancy of CRX alone is sufficient to encode silencers while the addition of a single NRL motif converts synthetic silencers to enhancers, and that genomic sequences with very high CRX motif content repress a basal promoter that lacks NRL motifs (*White et al., 2016*). We found that photoreceptor genes which are de-repressed upon loss of CRX are located near *cis*-regulatory sequences with high CRX motif content, and that genes near regions that are bound only by CRX are expressed at lower levels than genes near regions co-bound by CRX and NRL (*White et al., 2016*). In the current study, we find that silencers in our MPRA library are more likely to occur near de-repressed photoreceptor genes, while strong enhancers are enriched near genes that lose expression in *Crx*^-/- retina. These findings suggest that the low TF diversity and high CRX motif content that characterize silencers in our MPRA library are also important for silencing in the genome.

The contrast in motif diversity between enhancers and silencers that we observe could explain how CRX achieves selective activation and repression of its target genes in multiple cell types and across developmental time points (*Murphy et al., 2019*; *Ruzycki et al., 2018*). CRX itself is required for silencing, and we previously showed that some silencers become active enhancers in *Crx*^-/- retina (*White et al., 2016*). The mechanism of CRX-based silencing is unknown, however CRX cooperates with other TFs that can sometimes act as repressors of cell type-specific genes (*Chen et al., 2005*; *Peng et al., 2005*; *Webber et al., 2008*), while other repressors can directly inhibit activation by CRX or its co-activators (*Dorval et al., 2006*; *Hlawatsch et al., 2013*; *Mitton et al., 2003*; *Sanuki et al., 2010*). In *Drosophila* photoreceptors, selective silencing of opsin genes is controlled by cell type-specific expression of a repressor, Dve, which acts on the same K50 homeodomain-binding sites as a universally expressed activator, Otd, a homolog of CRX (*Rister et al., 2015*). Other transcriptional activators selectively act as repressors in the same cell type. GATA-1 represses the *GATA*-2 promoter by displacing CREB-binding protein (CBP), while at other genes GATA-1 binds CBP to activate transcription (*Grass et al., 2003*). Selective repression by GATA-1 is also mediated by chromatin occupancy levels and interaction with co-regulators (*Johnson et al., 2006*), which is consistent with our finding that sequence context enables a TF to both activate and repress genes in the same cell type.

Given the central role of CRX in selectively regulating genes in multiple closely related cell types (*Murphy et al., 2019*), we speculate that CRX-targeted silencers may contain sufficient information to act as enhancers in other cell types in which a different set of co-activating TFs are expressed. This hypothesis would be consistent with the finding that many silencers are enhancers in other cell types (*Doni Jayavelu et al., 2020*; *Gisselbrecht et al., 2020*; *Ngan et al., 2020*). Our work suggests that characterizing sequences by their motif information content offers a way to identify these different classes of *cis*-regulatory sequences in the genome.

# Materials and methods

## Key resources table

| Reagent type (species) or resource | Designation | Source or reference | Identifiers | Additional information |
|---|---|---|---|---|
| Strain, strain background (*Mus musculus*, male and female) | CD-1 | Charles River | Strain code 022 | |
| Recombinant DNA reagent | Library1 | This paper | | Listed in *Supplementary file 1* |
| Recombinant DNA reagent | Library2 | This paper | | Listed in *Supplementary file 2* |
| Recombinant DNA reagent | pJK01_Rhominprox-DsRed | *Kwasnieski et al., 2012* | AddGene plasmid # 173,489 | |
| Recombinant DNA reagent | pJK03_*Rho_basal*_DsRed | *Kwasnieski et al., 2012* | AddGene plasmid # 173,490 | |
| Sequence-based reagent | Primers | IDT | | Listed in *Supplementary file 6* |
| Commercial assay or kit | Monarch PCR Cleanup Kit | New England Biolabs | T1030S | |
| Commercial assay or kit | Monarch DNA Gel Extraction Kit | New England Biolabs | T1020L | |
| Commercial assay or kit | TURBO DNA-free | Invitrogen | AM1907 | |
| Commercial assay or kit | SuperScript III Reverse Transcriptase | Invitrogen | 18080044 | |
| Software, algorithm | Bedtools | https://bedtools.readthedocs.io/en/latest/ | RRID:SCR_006646 | |
| Software, algorithm | MEME Suite | https://meme-suite.org/ | RRID:SCR_001783 | |
| Software, algorithm | ShapeMF | https://github.com/h-samee/shape-motif, *Samee, 2021* | DOI:10.1016/j.cels.2018.12.001 | |
| Software, algorithm | Numpy | https://numpy.org/ | DOI:10.1038/s41586-020-2649-2 | |
| Software, algorithm | Scipy | https://www.scipy.org/ | DOI:10.1038/s41592-019-0686-2 | |
| Software, algorithm | Pandas | https://pandas.pydata.org/ | DOI:10.5281/zenodo.3509134 | |
| Software, algorithm | Matplotlib | https://matplotlib.org/ | DOI:10.5281/zenodo.1482099 | |
| Software, algorithm | Logomaker | https://github.com/jbkinney/logomaker, *Justin, 2021* | DOI:10.1093/bioinformatics/btz921 | |

## Library design

CRX ChIP-seq peaks re-processed by *Ruzycki et al., 2018* were intersected with previously published CRX MPRA libraries (*Hughes et al., 2018*; *White et al., 2013*) and one unpublished library to select sequences that had not been previously tested by MPRA. These sequences were scanned for instances of CRX motifs using FIMO version 4.11.2 (*Bailey et al., 2009*), a p-value cutoff of $2.3 \times 10^{-3}$ (see below), and a CRX PWM derived from an electrophoretic mobility shift assay (*Lee et al., 2010*). We centered 2622 sequences on the highest scoring CRX motif. For 677 sequences without a CRX motif, we instead centered them using the Gibbs sampler from ShapeMF (Github commit abe8421) (*Samee et al., 2019*) and a motif size of 10.

For sequences unbound in CRX ChIP-seq but in open chromatin, we took ATAC-seq peaks collected in 8 -week FACS-purified rods, green cones, and $Nrl^{-/-}$ blue cones (*Hughes et al., 2017*) and removed sequences that overlapped with CRX ChIP-seq peaks. The remaining sequences were scanned for

instances of CRX motifs using FIMO with a p-value cutoff of $2.5 \times 10^{-3}$ and the CRX PWM. Sequences with a CRX motif were kept and the three ATAC-seq data sets were merged together, intersected with H3K27ac and H3K4me3 ChIP-seq peaks collected in P14 retinas (*Ruzycki et al., 2018*), and centered on the highest scoring CRX motifs. We randomly selected 1004 H3K27ac$^+$H3K4me3$^-$ sequences and 541 H3K27ac$^+$H3K4me3$^+$ to reflect the fact that ~35 % of CRX ChIP-seq peaks are H3K4me3$^+$. After synthesis of our library, we discovered 11 % of these sequences do not actually overlap H3K27ac ChIP-seq peaks (110/1004 of the H3K4me3$^-$ group and 60/541 of the H3K4me3$^+$ group), but we still included them in the analysis because they contain CRX motifs in ATAC-seq peaks.

All data was converted to mm10 coordinates using the UCSC liftOver tool (*Haeussler et al., 2019*) and processed using Bedtools version 2.27.1 (*Quinlan and Hall, 2010*). All sequences in our library design were adjusted to 164 bp and screened for instances of EcoRI, SpeI, SphI, and NotI sites. In total, our library contains 4844 genomic sequences (2622 CRX ChIP-seq peaks with motifs, 677 CRX ChIP-seq peaks without motifs, 1004 CRX$^-$ATAC$^+$H3K27ac$^+$H3K4me3$^-$ CRX motifs, and 541 CRX$^-$ATAC$^+$H3K27ac$^+$H3K4me3$^+$ CRX motifs), a variant of each sequence with all CRX motifs mutated, 150 scrambled sequences, and a construct for cloning the basal promoter alone.

For sequences centered on CRX motifs, all CRX motifs with a p-value of $2.5 \times 10^{-3}$ or less were mutated by changing the core TAAT to TACT (*Lee et al., 2010*) on the appropriate strand, as described previously (*Hughes et al., 2018*; *White et al., 2013*). We then re-scanned sequences and mutated any additional motifs inadvertently created.

To generate scrambled sequences, we randomly selected 150 CRX ChIP-seq peaks spanning the entire range of GC content in the library. We then scrambled each sequence while preserving dinucleotide content as previously described (*White et al., 2013*). We used FIMO to confirm that none of the scrambled sequences contain CRX motifs.

We unintentionally used a FIMO p-value cutoff of $2.3 \times 10^{-3}$ to identify CRX motifs in CRX ChIP-seq peaks, rather than the slightly less stringent $2.5 \times 10^{-3}$ cutoff used with ATAC-seq peaks or mutating CRX motifs. Due to this anomaly, there may be sequences centered using ShapeMF that should have been centered on a CRX motif, and these motifs would not have been mutated because CRX motifs were not mutated in sequences centered using ShapeMF. However, any intact CRX motifs would still be captured in the residual information content of the mutant sequence.

## Plasmid library construction

We generated two 15,000 libraries of 230 bp oligonucleotides (oligos) from Agilent Technologies (Santa Clara, CA) through a limited licensing agreement. Our library was split across the two oligo pools, ensuring that both the genomic and mutant forms of each sequence were placed in the same oligo pool (*Supplementary files 1 and 2*). Both oligo pools contain all 150 scrambled sequences as an internal control. All sequences were assigned three unique barcodes as previously described (*White et al., 2013*). In each oligo pool, the basal promoter alone was assigned 18 unique barcodes. Oligos were synthesized as follows: 5′ priming sequence (GTAGCGTCTGTCCGT)/EcoRI site/Library sequence/SpeI site/C/SphI site/Barcode sequence/NotI site/3′ priming sequence (CAACTACTACTACAG). To clone the basal promoter into barcoded oligos without any upstream *cis*-regulatory sequence, we placed the SpeI site next to the EcoRI site, which allowed us to place the promoter between the EcoRI site and the 3′ barcode.

We cloned the synthesized oligos as previously described by our group (*Kwasnieski et al., 2012White et al., 2016*; *White et al., 2013*). Specifically, for each oligo pool, we used 50 femtomoles of template and four cycles of PCR in each of multiple 50 µl reactions (New England Biolabs [NEB], Ipswich, MA) (NEB Phusion) using primers MO563 and MO564 (*Supplementary file 6*), 2 % DMSO, and an annealing temperature of 57 °C. PCR amplicons were purified from a 2 % agarose gel (NEB), digested with EcoRI-HF and NotI-HF (NEB), and then cloned into the EagI and EcoRI sites of the plasmid pJK03 with multiple 20 µl ligation reactions (NEB T4 ligase). The libraries were transformed into 5-alpha electrocompetent cells (NEB) and grown in liquid culture. Next, 2 µg of each library was digested with SphI-HF and SpeI-HF (NEB) and then treated with Antarctic phosphatase (NEB).

The *Rho* basal promoter and *DsRed* reporter gene was amplified from the plasmid pJK01 using primers MO566 and MO567 (*Supplementary file 6*). The Polylinker and *DsRed* reporter gene was amplified from the plasmid pJK03 using primers MO610 and MO567 (*Supplementary file 6*). The Polylinker is a short 23 bp multiple cloning site with no known core promoter motifs. Inserts were

purified from a 1 % agarose gel (NEB), digested with NheI-HF and SphI-HF (NEB), and cloned into the libraries using multiple 20 µl ligations (NEB T4 ligase). The libraries were transformed into 5-alpha electrocompetent cells (NEB) and grown in liquid culture.

## Retinal explant electroporation

Animal procedures were performed in accordance with a Washington University in St Louis Institutional Animal Care and Use Committee-approved vertebrate animals protocol. Electroporation into retinal explants and RNA extraction was performed as described previously (*Hsiau et al., 2007*; *Hughes et al., 2018*; *Kwasnieski et al., 2012*; *White et al., 2016*; *White et al., 2013*). Briefly, retinas were isolated from P0 newborn CD-1 mice and electroporated in a solution with 30 µg library and 30 µg *Rho*-GFP. Electroporated retinas were cultured for 8 days, at which point they were harvested, washed three times with HBSS (ThermoFisher Scientific/Gibco, Waltham, MA), and stored in TRIzol (ThermoFisher Scientific/Invitrogen, Waltham, MA) at –80 °C. Five retinas were pooled for each biological replicate and three replicates were performed for each library. RNA was extracted from TRIzol according to manufacturer's instructions and treated with TURBO DNase (Invitrogen). cDNA was prepared using SuperScript RT III (Invitrogen) with oligo dT primers. Barcodes from both the cDNA and the plasmid DNA pool were amplified for sequencing (described below). The resulting products were mixed at equal concentration and sequenced on the Illumina NextSeq platform. We obtained greater than 1300 × coverage across all samples.

*Rho* libraries were amplified using primers MO574 and MO575 (*Supplementary file 6*) for six cycles at an annealing temperature of 66 °C followed by 18 cycles with no annealing step (NEB Phusion) and then purified with the Monarch PCR kit (NEB). PCR amplicons were digested using MfeI-HF and SphI-HF (NEB) and ligated to custom-annealed adaptors with PE2 indexing barcodes and phased P1 barcodes (*Supplementary file 6*). The final enrichment PCR used primers MO588 and MO589 (*Supplementary file 6*) for 20 cycles at an annealing temperature of 66 °C (NEB Phusion), followed by purification with the Monarch PCR kit. Polylinker libraries were amplified using primers BC_CRX_Nested_F and BC_CRX_R (*Supplementary file 6*) for 30 cycles (NEB Q5) at an annealing temperature of 67 °C and then purified with the Monarch PCR kit. Illumina adaptors were then added via two further rounds of PCR. First, P1 indexing barcodes were added using forward primers P1_inner_A through P1_inner_D and reverse primer P1_inner_nested_rev (*Supplementary file 6*) for five cycles at an annealing temperature of 55 °C followed by five cycles with no annealing step (NEB Q5). PE2 indexing barcodes were then added by amplifying 2 µl of the previous reaction with forward primer P1_outer and reverse primers PE2_outer_SIC69 and PE2_outer_SIC70 (*Supplementary file 6*) for five cycles at an annealing temperature of 66 °C followed by five cycles with no annealing step (NEB Q5) and then purified with the Monarch PCR kit.

## Data processing

All data processing, statistical analysis, and downstream analyses were performed in Python version 3.6.5 using Numpy version 1.15.4 (*Harris et al., 2020*), Scipy version 1.1.0 (*Virtanen et al., 2020*), and Pandas version 0.23.4 (*McKinney, 2010*), and visualized using Matplotlib version 3.0.2 (*Hunter, 2007*) and Logomaker version 0.8 (*Tareen and Kinney, 2020*). All statistical analysis used two-sided tests unless noted otherwise.

Sequencing reads were filtered to ensure that the barcode sequence perfectly matched the expected sequence (>93% reads in a sample for the *Rho* libraries, >86% reads for the Polylinker libraries). For the *Rho* libraries, barcodes that had less than 10 raw counts in the DNA sample were considered missing and removed from downstream analysis. Barcodes that had less than five raw counts in any cDNA sample were considered present in the input plasmid pool but below the detection limit and thus set to zero in all samples. Barcode counts were normalized by reads per million (RPM) for each sample. Barcode expression was calculated by dividing the cDNA RPM by the DNA RPM. Replicate-specific expression was calculated by averaging the barcodes corresponding to each library sequence. After performing statistical analysis (see below), expression levels were normalized by replicate-specific basal mean expression and then averaged across biological replicates.

For the Polylinker assay, the expected lack of expression of many constructs required different processing. Barcodes that had less than 50 raw counts in the DNA sample were removed from downstream analysis. Barcodes were normalized by RPM for each replicate. Barcodes that had less

than 8 RPM in any cDNA sample were set to zero in all samples. cDNA RPM were then divided by DNA RPM as above. Within each biological replicate, barcodes were averaged as above but were not normalized to basal expression because there is no basal construct. Expression values were then averaged across biological replicates. Due to the low expression of scrambled sequences and the lack of a basal construct, we were unable to assess data calibration with the same rigor as above.

## Assignment of activity classes

Activity classes were assigned by comparing expression levels to basal promoter expression levels across replicates. The null hypothesis is that the expression of a sequence is the same as basal levels. Expression levels were approximately log-normally distributed, so we computed the log-normal parameters for each sequence and then performed Welch's t-test. We corrected for multiple hypotheses using the Benjamini-Hochberg FDR procedure. We corrected for multiple hypotheses in each library separately to account for any potential batch effects between libraries. The $\log_2$ expression was calculated after adding a pseudocount of $1 \times 10^{-3}$ to every sequence.

Sequences were classified as enhancers if they were twofold above basal and the q-value was below 0.05. Silencers were similarly defined as twofold below basal and q-value less than 0.05. Inactive sequences were defined as within a twofold change and q-value greater than or equal to 0.05. All other sequences were classified as ambiguous and removed from further analysis. We used scrambled sequences to further stratify enhancers into strong and weak enhancers, using the rationale that scrambled sequences give an empirical distribution for the activity of random sequences. We defined strong enhancers as enhancers that are above the 95th percentile of scrambled sequences and all other enhancers as weak enhancers.

For the Polylinker assay, we did not have a basal construct as a reference point. Instead, we defined a sequence to have autonomous activity if the average cDNA barcode counts were higher than average DNA barcode counts, and non-autonomous otherwise. The $\log_2$ expression was calculated after adding a pseudocount of $1 \times 10^{-2}$ to every sequence.

## RNA-seq analysis

We obtained RNA-seq data from WT and Crx$^{-/-}$ P21 retinas (*Roger et al., 2014*) processed into a counts matrix for each gene by *Ruzycki et al., 2018*. Each sample was normalized by read counts per million and replicates were averaged together. Genes with at least a twofold change between genotypes were considered differentially expressed. We determined which differentially expressed genes are near a member of our library using previously published associations between retinal ATAC-seq peaks and genes (*Murphy et al., 2019*). For de-repressed genes, we determined how often the nearest library member is a silencer; for down-regulated genes, we determined how often the nearest library member is a strong or weak enhancer.

## Motif analysis

We performed motif enrichment analysis using the MEME Suite version 5.0.4 (*Bailey et al., 2009*). We searched for motifs that were enriched in one group of sequences relative to another group using DREME-py3 with the parameters -mink 6 -maxk 12 -e 0.05 and compared the de novo motifs to known motifs using TOMTOM on default parameters. We ran DREME using strong enhancers as positives and silencers as negatives, and vice versa. For TOMTOM, we used version 11 of the full mouse HOCOMOCO database (*Kulakovskiy et al., 2018*) with the following additions from the JASPAR human database (*Khan et al., 2018*): NRL (accession MA0842.1), RORB (accession MA1150.1), and RAX (accession MA0718.1). We added these PWMs because they have known roles in the retina, but the mouse PWMs were not in the HOCOMOCO database. We also used the CRX PWM that we used to design the library. Motifs were selected for downstream analysis based on their matches to the de novo motifs, whether the TF had a known role in retinal development, and the quality of the PWM. Because PWMs from TFs of the same family were so similar, we used one TF for each DREME motif, recognizing that these motifs may be bound by other TFs that recognize similar motifs. We did not use any PWMs with a quality of 'D'. We excluded DREME motifs without a match to the database from further analysis; most of these resemble dinucleotides.

## Predicted occupancy

We computed predicted occupancy as previously described (**White et al., 2013**; **Zhao et al., 2009**). Briefly, we normalized each letter probability matrix by the most probable letter at each position. We took the negative log of this matrix and multiplied by 2.5, which corresponds to the ideal gas constant times 300 K, to obtain an energy weight matrix. We used a chemical potential $\mu$ of 9 for all TFs. At this value, the probability of a site being bound is at least 0.5 if the relative $K_D$ is at least 0.03 of the optimal binding site. We computed the predicted occupancy for every site on the forward and reverse strands and summed them together to get a single value for each TF.

To determine if there is a bias in the linear arrangement of motifs, we selected strong enhancers with exactly one site occupied by CRX and exactly one site occupied by a second TF. We counted the number of times the position of the second TF was 5′ and 3′ of the CRX site and then performed a binomial test. We did not observe a statistically significant bias for any TF at an FDR q-value cutoff of 0.05. We also performed this analysis for silencers with exactly one site occupied by CRX and exactly one site occupied by NRL and did not observe a significant difference in the 5′ vs. 3′ bias of strong enhancers vs. silencers (Fisher's exact test p = 0.17).

## Information content

To capture the effects of TF predicted occupancy and diversity in a single metric, we calculated the motif information content using Boltzmann entropy. Boltzmann's equation states that the entropy of a system $S$ is related to the number of ways the molecules can be arranged (microstates) $W$ via the equation $S = k_B \log W$, where $k_B$ is Boltzmann's constant (**Phillips et al., 2012**, Chapter 5). The number of microstates is defined as $W = \frac{N!}{\prod_i N_i!}$ where $N$ is the total number of particles and $N_i$ are the number of the -th type of particles. In our case, the system is the collection of predicted binding motifs for different TFs in a *cis*-regulatory sequence. We assume each TF is a different type of molecule because the DNA-binding domain of each TF belongs to a different subfamily. The number of molecular arrangements $W$ represents the number of distinguishable ways that the TFs can be ordered on the sequence. Thus, $N_i$ is the predicted occupancy of the -th TF and $N$ is the total predicted occupancy of all TFs on the *cis*-regulatory sequence. Because the predicted occupancies are continuous values, we exploit the definition of the Gamma function, $\Gamma(N + 1) = N!$ to rewrite $W = \frac{\Gamma(N+1)}{\prod_i \Gamma(N_i+1)}$ .

If we assume that each arrangement of motifs is equally likely, then we can write the probability of arrangement $w = 1, \ldots, W$ as $p_w = \frac{1}{w}$ and rewrite the entropy as $S = -\log\left(\frac{1}{w}\right) = -\log(p_w)$, where we have dropped Boltzmann's constant since the connection between molecular arrangements and temperature is not important. Because each arrangement is equally likely, then $\frac{1}{w}$ is also the expected value of $p_w$ and we can write the entropy as $S = -E[\log(p_w)] = -\sum_w p_w \log(p_w)$ , which is Shannon entropy. By definition, Shannon entropy is also the expected value of the information content: $E[I] = -\sum_w p_w \log(p_w) = \sum_w p_w I(w)$ where the information content $I$ of a particular state is $I(w) = \log(p_w)$. Since we assumed each arrangement is equally likely, then the expected value of the information content is also the information content of each arrangement. Therefore, the information content of a *cis*-regulatory sequence can be written as $I = -\log(p_w) = \log W$. We use log base 2 to express the information content in bits.

With this metric, *cis*-regulatory sequences with higher predicted TF occupancies generally have higher information content. Sequences with higher TF diversity have higher information content than lower diversity sequences with the same predicted occupancy. Thus, our metric captures the effects of both TF diversity and total TF occupancy. For example, consider hypothetical TFs A, B, and C. If motifs for only one TF are in a sequence, then $W$ is always one and the information content is always zero (regardless of total occupancy). The simplest case for non-zero information content is one motif for A, one motif for B, and zero motifs for C (1-1-0). Then $W = \frac{2!}{1!1!} = 2$ and $I = 1$ bit. If we increase predicted occupancy by adding a motif for A (2-1-0), then $W = \frac{3!}{2!1!} = 3$ and $I = 1.6$ bits, which is approximately the information content of silencers and inactive sequences. If we increase predicted occupancy again and add a second motif for B (2-2-0), then $W = \frac{4!}{2!2!} = 6$ and $I = 2.6$ bits, which is approximately the information content of strong enhancers. If instead of increasing predicted occupancy, we instead increase diversity by replacing a motif for A with a motif for C (1-1-1), then $W = \frac{3!}{1!1!1!} = 6$ and once again $I = 2.6$ bits, which is higher than the lower diversity case (2-1-0).

According to **Wunderlich and Mirny, 2009**, the probability of observing $k$ total motifs for $m$ different TFs in a $w$ bp window is $p(k) \sim (Poisson(k; \lambda))$, where $\lambda = pmw$ and $p$ is the probability of

finding a spurious motif in the genome. The expected number of windows with $k$ total motifs in a genome of length is thus $E(k) = p(k) \cdot N$. In mammals, $N \approx 10^9$ and Wunderlich and Mirny find that $p = 0.0025$ for multicellular eukaryotes. For $m = 3$ TFs and a bp window (which is the size of our sequences), $\lambda = 0.123$ and $E(5) = 1.6$ meaning that five total motifs for three different TFs specify an approximately unique 164 bp location in a mammalian genome. Five total motifs for three different TFs can be achieved in two ways: three motifs for A, one for B, and one for C (3-1-1), or two motifs for A, two for B, and one for C (2-2-1). In the case of 3-1-1, $W = \frac{5!}{3!1!1!} = 20$ and $I = 4.3$ bits. In the case of 2-2-1, $W = \frac{5!}{2!2!1!} = 30$ and $I = 4.9$ bits.

## Machine learning

The $k$-mer SVM was fit using gkmSVM (*Ghandi et al., 2014*). All other machine learning, including cross-validation, logistic regression, and computing ROC and PR curves, was performed using scikit-learn version 0.19.1 (*Pedregosa et al., 2011*). We wrote custom Python wrappers for gkmSVM to allow for interfacing between the C++ binaries and the rest of our workflow. We ran gkmSVM with the parameters -l 6 -k 6 -m 1. To estimate model performance, all models were fit with stratified fivefold cross-validation after shuffling the order of sequences. For the TF occupancy logistic regression model, we used L2 regularization. We selected the regularization parameter C by performing grid search with fivefold cross-validation on the values $10^{-4}$, $10^{-3}$, $10^{-2}$, $10^{-1}$, 1, $10^1$, $10^2$, $10^3$, $10^4$ and selecting the value that maximized the F1 score. The optimal value of C was 0.01, which we used as the regularization strength when assessing the performance of the model with other feature sets.

To assess the performance of the logistic regression model, we randomly sampled eight PWMs from the HOCOMOCO database and computed the predicted occupancy of each TF on each sequence. We then fit a new logistic regression model with these features and repeated this procedure 100 times to generate a background distribution of model performances.

To generate de novo motifs from the SVM, we generated all 6-mers and scored them against the SVM. We then ran the svmw_emalign.py script from gkmSVM on the $k$-mer scores with the parameters -n 10 -f 2 -m 4 and a PWM length of 6, and then used TOMTOM to compare them to the database from our motif analysis.

## Other data sources

We used our previously published library (*White et al., 2013*) as an independent test set for our machine learning models. We defined strong enhancers as ChIP-seq peaks that were above the 95th percentile of all scrambled sequences. There was no basal promoter construct in this library, so instead we defined silencers as ChIP-seq peaks that were at least twofold below the $\log_2$ mean of all scrambled sequences.

Previously published ChIP-seq data for NRL (*Hao et al., 2012*) that was re-processed by *Hughes et al., 2017* and MEF2D (*Andzelm et al., 2015*) was used to annotate sequences for in vivo TF binding. We converted peaks to mm10 coordinates using the UCSC liftOver tool and then used Bedtools to intersect peaks with our library.

## Acknowledgements

We thank Gary Stormo and members of the Cohen Lab for critically reading the manuscript and helpful discussions; Philip A Ruzycki, Andrew EO Hughes, and Timothy J Cherry for providing processed ChIP-seq and RNA-seq data; and Jessica Hoisington-Lopez and MariaLynn Crosby from the DNA Sequencing Innovation Lab for assistance with high-throughput sequencing.

## Additional information

### Funding

| Funder | Grant reference number | Author |
| --- | --- | --- |
| National Institutes of Health | F31HG011431 | Ryan Z Friedman |

| Funder | Grant reference number | Author |
| --- | --- | --- |
| National Institutes of Health | R01GM121755 | Michael A White |
| National Institutes of Health | R01EY027784 | Barak A Cohen |
| National Institutes of Health | R01EY025196 | Joseph C Corbo |
| National Institutes of Health | R01EY030075 | Joseph C Corbo |

The funders had no role in study design, data collection and interpretation, or the decision to submit the work for publication.

## Author contributions

Ryan Z Friedman, Conceptualization, Data curation, Formal analysis, Funding acquisition, Investigation, Methodology, Software, Visualization, Writing – original draft, Writing – review and editing; David M Granas, Connie A Myers, Investigation; Joseph C Corbo, Funding acquisition, Supervision, Writing – original draft, Writing – review and editing; Barak A Cohen, Conceptualization, Funding acquisition, Methodology, Supervision, Writing – original draft, Writing – review and editing; Michael A White, Conceptualization, Funding acquisition, Supervision, Writing – original draft, Writing – review and editing

## Author ORCIDs

Ryan Z Friedman http://orcid.org/0000-0001-9013-8676
Joseph C Corbo http://orcid.org/0000-0002-9323-7140
Barak A Cohen http://orcid.org/0000-0002-3350-2715
Michael A White http://orcid.org/0000-0001-8511-6026

## Ethics

This study was performed in strict accordance with the recommendations in the Guide for the Care and Use of Laboratory Animals of the National Institutes of Health. All of the animals were handled according to protocol # A-3381-01 approved by the Institutional Animal Care and Use Committee of Washington University in St. Louis. Euthanasia of mice was performed according to the recommendations of the American Veterinary Medical Association Guidelines on Euthanasia. Appropriate measures are taken to minimize pain and discomfort to the animals during experimental procedures.

## Decision letter and Author response

Decision letter https://doi.org/10.7554/eLife.67403.sa1
Author response https://doi.org/10.7554/eLife.67403.sa2

# Additional files

## Supplementary files

• Supplementary file 1. FASTA file of all sequences in library 1. Sequences were named using the following nomenclature: 'chrom-start-stop_annotations_variant'. 'Chrom', 'start', and 'stop' correspond to the mm10 genomic coordinates of the sequences in BED format. 'Annotations' is a four-letter string where the first position indicates CRX-binding status (ChIP-seq peak or Unbound), the second position indicates CRX motif status (PWM hit, Shape motif, or Both PWM and shape motif), the third position indicates ATAC-seq status (peak in Rods but not cones, peak in Cones but not rods, peak in both rod and cone Photoreceptors, or peak in None of the above), and the fourth position indicates histone ChIP-seq status ('Enhancer marked' with H3K27Ac$^+$H3K4me3$^-$, 'Promoter marked' with H3K27Ac$^+$H3K4me3$^+$, Q for H3K27Ac$^-$H3K4me3$^+$, or Neither mark). 'Variant' indicates whether the sequence is genomic ('WT'), mutated CRX motifs ('MUT-allCrxSites'), scrambled shape motif ('MUT-shape'), or a scrambled control ('scrambled').

• Supplementary file 2. FASTA file of all sequences in library 2. Sequences were named as in *Supplementary file 1*.

• Supplementary file 3. Expression measurements and annotations of all sequences. Values are tab-delimited. Rows are named based on the sequence name from *Supplementary files 1 and 2*

without the 'variant' information. Columns ending in '_WT' indicate the wild-type sequence with the *Rho* promoter, '_MUT' as the CRX motif mutant sequence with the *Rho* promoter, and '_POLY' as the wild-type sequence with the Polylinker. Sequences with the scrambled shape motif were excluded from the '_MUT' columns. Columns are named as follows: label, the sequence name from *Supplementary files 1 and 2* without the 'variant' information; expression, average activity of the sequence, NaN indicates sequence was missing from the plasmid pool; expression_std, standard deviation of activity; expression_reps, number of replicates in which the sequence was measured; expression_pvalue, p-value from Welch's t-test of log-normal data for the null hypothesis that the activity of the sequence with *Rho* is no different than the *Rho* promoter alone; expression_qvalue, FDR-correction of the p-values; library, which library contains the sequence; expression_log2, log2 average activity of the sequence; group_name, activity classification of the sequence with the *Rho* promoter; plot_color, hex code for visualization; variant, the 'variant' portion of the sequence identifier; wt_vs_mut_log2, log2 fold change between the wild-type and mutant version of the sequence, NaN indicates the wild-type and/or mutant version was not measured; wt_vs_mut_pvalue, p-value from Welch's t-test for the null hypothesis that the wild-type and mutant sequences have the same activity; wt_vs_mut_qvalue, FDR-correction of the p-values; autonomous_activity, Boolean value for if the wild-type sequence is autonomous with the Polylinker; crx_bound, nrl_bound, and mef2d_bound, Boolean values for if the sequence overlaps a ChIP-seq peak for the corresponding TF; binding_group, string denoting each of the eight possible combinations of CRX, NRL, and MEF2D binding.

• Supplementary file 4. Predicted occupancy scores for each transcription factor (TF) and each sequence. Values are tab-delimited. Rows are named based on the sequence name from *Supplementary files 1 and 2* including the 'variant' information. Columns are the predicted occupancy scores for the denoted TF.

• Supplementary file 5. Information content and related metrics for each sequence. Values are tab-delimited. Rows are named based on the sequence name from *Supplementary files 1 and 2*, including the 'variant' information. Columns are named as follows: total_occupancy, total predicted occupancy of all eight transcription factors (TFs); diversity, number of TFs with predicted occupancy above 0.5; entropy, information content (which is also entropy).

• Supplementary file 6. Primers used in this study.

• Transparent reporting form

## Data availability

The pJK01 and pJK03 plasmids have been deposited with AddGene (IDs 173489, 173490). Raw sequencing data and barcode counts have been uploaded to the NCBI GEO database under accession GSE165812. All processed activity data, predicted occupancy, and information content values are available in the supplementary material. All code for data processing, analysis, and visualization is available on Github at https://github.com/barakcohenlab/CRX-Information-Content (copy archived at https://archive.softwareheritage.org/swh:1:rev:ca108a6fb1d30c9476521eeb7e77f921a4c99323).

The following dataset was generated:

| Author(s) | Year | Dataset title | Dataset URL | Database and Identifier |
|---|---|---|---|---|
| Friedman RZ, Granas DM, Myers CA, Corbo JC, Cohen BA, White MA | 2021 | Information Content Differentiates Enhancers From Silencers in Mouse Photoreceptors | https://www.ncbi.nlm.nih.gov/geo/query/acc.cgi?acc=GSE165812 | NCBI Gene Expression Omnibus, GSE165812 |

The following previously published datasets were used:

| Author(s) | Year | Dataset title | Dataset URL | Database and Identifier |
|---|---|---|---|---|
| Langmann T, Corbo JC | 2010 | Deciphering the cis-regulatory architecture of mammalian photoreceptors | https://www.ncbi.nlm.nih.gov/geo/query/acc.cgi?acc=GSE20012 | NCBI Gene Expression Omnibus, GSE20012 |

*Continued on next page*

*Continued*

| Author(s) | Year | Dataset title | Dataset URL | Database and Identifier |
|---|---|---|---|---|
| Hughes AE, Enright JM, Myers CA, Shen SQ, Corbo JC | 2016 | ATAC-seq and RNA-seq of adult mouse rods and cones | https://www.ncbi.nlm.nih.gov/geo/query/acc.cgi?acc=GSE83312 | NCBI Gene Expression Omnibus, GSE83312 |
| Ruzycki PA, Zhang X, Chen S | 2018 | CRX directs photoreceptor differentiation by accelerating chromatin remodeling at specific target sites | https://static-content.springer.com/esm/art%3A10.1186%2Fs13072-018-0212-2/MediaObjects/13072_2018_212_MOESM1_ESM.xlsx | Additional file 1, static-content.springer.com/esm/art%3A10.1186%2Fs13072-018-0212-2/MediaObjects/13072_2018_212_MOESM1_ESM.xlsx |
| Hao H, Kim DS, Klocke B, Johnson KR, Cui K, Gotoh N, Zang C, Gregorski J, Gieser L, Peng W, Fann Y, Seifert M, Zhao K, Swaroop A | 2012 | Transcriptional Regulation of Rod Photoreceptor Homeostasis Revealed by In Vivo NRL Targetome Analysis | https://datashare.nei.nih.gov/nnrlMain.jsp | NEI Data Share, Hong PLoS-Genet-2012 |
| Andzelm MM, Cherry TJ, Harmin DA, Boeke AC, Lee C, Hemberg M, Pawlyk B, Malik AN, Flavell SW, Sandberg MA, Raviola E, Greenberg ME | 2015 | MEF2D drives photoreceptor development through a genome-wide competition for tissue-specific enhancers | https://www.ncbi.nlm.nih.gov/geo/query/acc.cgi?acc=GSE61392 | NCBI Gene Expression Omnibus, GSE61392 |

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
