## [Decision Letter]

**Acceptance summary:**

This manuscript will be of interest to geneticists seeking to establish rules that govern gene regulation. To explain why a sequence enhances, rather than silences, gene transcription the authors draw our attention away from the binding of a single transcription factor, to focus instead on the number and diversity of transcription factor molecules that bind to it. Using a relatively simple metric called sequence information content they appear to be able to improve the prediction of enhancer over silencer sequences. A concern is whether the silencers are true silencers, or whether they only act as such in this specific experimental paradigm.

**Decision letter after peer review:**

[Editors’ note: the authors submitted for reconsideration following the decision after peer review. What follows is the decision letter after the first round of review.]

Thank you for submitting your work entitled "Information content differentiates enhancers from silencers in mouse photoreceptors" for consideration by *eLife*. Your article has been reviewed by 2 peer reviewers, and the evaluation has been overseen by a Reviewing Editor and a Senior Editor. The following individual involved in review of your submission has agreed to reveal their identity: Chris P Ponting (Reviewer #1).

Based on the reviews we received, I am sorry that we cannot offer publication in *eLife*.

While one of the reviewers was very positive, the other reviewer pointed out substantial weaknesses. We acknowledge that your title clearly makes narrow claims, but for *eLife* we would have hoped that the findings could be generalized.

*Reviewer #1:*

Why can silencers be enhancers in other cell types? Why is it that active chromatin epigenetic marks or binding of a single transcription factor do not reliably predict active enhancers? These are thorny issues in genomics because they hinder our mechanistic understanding of gene transcription regulation.

In this well-written submission the authors go beyond their previous publications using the same experimental system (White et al., 2013, 2016). They use MPRA for the CRX transcription factor (TF) in explanted mouse retinas to show that epigenetically indistinguishable sequences are classified more accurately as enhancers or silencers by the number and diversity of lineage-specific transcription factor binding motifs that they contain. They separate enhancers from silencers by enhancers' more diverse collection of TF motifs. This distinction is captured in a metric called sequence information content calculated from both TF motif count and diversity. This single metric is slightly worse at predicting strong enhancers over silencers than a model considering the PWMs for 8 TFs.

1. Whether the authors observe a bias in the linear arrangement of these TFs' motifs that might assist in distinguishing enhancers from silencers?

2. p10 the choice of the 8 lineage-defining TFs was somewhat arbitrary because of the arbitrary nature of PWM significance thresholds. Please justify their choice and number, and comment on how well the model performs when this TF set is altered?

Does an evolutionary change in information content calculated between orthologous *Mus musculus* and, say, Mus spretus sequence help to separate active (enhancer/silencer) sequence from inactive sequence? (https://doi.org/10.1038/sdata.2016.75).

p9 please explain further "we chose to represent the zinc finger motif with MAZ based on the PWM qualities".

p21 why were different FIMO p-value thresholds applied?

p27 line 562 "silencers as negatives"?

line 132 "For motifs that matched multiple TFs, we selected one representative TF for downstream analysis (Figure 2—figure supplement 2, Methods)". Please explain further.

*Reviewer #2:*

The authors state that enhancers and silencers often have the same epigenomic profiles and attempt to identify sequence-based information to differentiate between the two types of elements. They use massively parallel reporter assays to test elements that bind CRX for activity in retinal explants. The authors then look for differences in motif content between the elements that act as silencers vs. those that act as enhancers of gene expression from a basal promoter. They find that although enhancers and silencers have motifs for the same transcription factor – CRX, the number of sites and diversity of other TF sites is greater within enhancers. They suggest motif content is a way to distinguish between the two types of elements. I'm not convinced that anything can be determined about silencers using this experimental design.

Strengths:

The authors test many putative enhancers in mouse retinas and identify elements whose function requires CRX sites.

Interestingly, different behaviors of functional elements could not be predicted based on differences in DNA accessibility or ATAC-seq peak or CRX occupancy. This is a nice systematic example of how difficult it is to predict an enhancer strength or activity based on differences in epigenomic data and highlights the need for sequence-based approaches to identify the specific activity of an element.

They do a nice analysis of the inert vs. weak and strong enhancers. The data and analysis of these experiments could be really informative for understanding why not all regions that bind CRX and are within open chromatin are active enhancers.

Weaknesses:

I'm concerned that the silencers they detect could be an artifact of the experimental design. The promoter contains CRX sites and NRL sites, so there is some level of basal expression; the silencers are enriched in repressors, so is it just that the elements containing a repressor are silencing the basal transcription? Moreover, what does this mean relative to the elements in the endogenous locus, if an endogenous promoter doesn't have CRX or NRL sites within its promoter or basal transcription does this mean the silencers as described in this assay are not really silencers within the genome. I don't think it is possible to make conclusions about a cis-reg element's silencer capacity based on these experiments.

In line with this, they find that the silencers bind CRX in combination with a repressive TF. Would they find that enhancers as they define them bind a combination of transcriptional activators and that silencers bind some activators such as CRX in combination with a transcriptional repressor expressed in the cell type where the element acts as a silencer?

I am also not convinced that silencers and enhancers are different things. If a genomic element controls the time and location of gene expression, then it is an enhancer – enhancers can bind activators and repressors and restrict expression to only particular cell types. I think trying to call things enhancers, and silencers makes things overly complex, especially considering the fact the authors point out that the same element can be an enhancer in one tissue type and a silencer in another. I am also concerned about this in relation to my previous comments on the experimental design and the issues demonstrating that a silencer really works this way within the genome.

As silencers are decreasing expression from a basal promoter and the whole paper is centered on this, I think the choice of promoter in this experiment is critical. I don't think one promoter can be used to draw such conclusions. The authors use a basal promoter for the Rho gene, which contains one NRL site and CRX sites, and these could work in combination with the sites within the elements being tested. I wonder if the elements they're testing would behave the same way with their endogenous promoters or with another promoter that does not contain CRX or NRL sites. Testing these elements with no promoter does not really address this question. It would be helpful to test these elements with the Rho promoter where you mutate the NRL and CRX sites. It would also be helpful to test these elements with a different promoter for another gene that is expressed in the retina, ideally on that doesn't contain CRX and NRL sites, to see if there are enhancer-promoter interactions that are influencing your results and thus your conclusions. If the endogenous promoters don't contain CRX or NRL sites or don't have a basal level of transcription, would your element really be a silencer? I don't think it would, and I'm concerned that the results you're seeing are an artifact of your experimental design.

The paper claims that information content can be used to distinguish between these two classes of element, I would like to see how this compares to prevalence to transcriptional repressors, and activators found in silencers and enhancers as the only TF mentioned to work with CRX in the silencer is a well-known repressor Snail. Would the presence of a repressive TF be a better predictor of silencer vs. enhancer activity?

I would like to see that the information content model performs better than measuring the prevalence of activator and repressor motifs.

In line with this, the difference in information content between a silencer and an enhancer is 3 motifs for 2 tfs vs. 3 motifs for 3 tfs or 4 motifs for 2 tfs. For the silencers and enhancers, what % of these TF motifs are repressors, and what % are activators.

In my opinion, the authors should focus on using their data to work out what makes regions under their genomic marks functional enhancers vs. inert elements.

---

## [Author Response]

[Editors’ note: The authors appealed the original decision. What follows is the authors’ response to the first round of review.]

Reviewer #1:Why can silencers be enhancers in other cell types? Why is it that active chromatin epigenetic marks or binding of a single transcription factor do not reliably predict active enhancers? These are thorny issues in genomics because they hinder our mechanistic understanding of gene transcription regulation.In this well-written submission the authors go beyond their previous publications using the same experimental system (White et al., 2013, 2016). They use MPRA for the CRX transcription factor (TF) in explanted mouse retinas to show that epigenetically indistinguishable sequences are classified more accurately as enhancers or silencers by the number and diversity of lineage-specific transcription factor binding motifs that they contain. They separate enhancers from silencers by enhancers' more diverse collection of TF motifs. This distinction is captured in a metric called sequence information content calculated from both TF motif count and diversity. This single metric is slightly worse at predicting strong enhancers over silencers than a model considering the PWMs for 8 TFs.1. Whether the authors observe a bias in the linear arrangement of these TFs' motifs that might assist in distinguishing enhancers from silencers?

We do not observe a bias: (1) A biased arrangement of non-CRX TFs would be evident as co-occurrence of TF pairs, which we did not observe as shown in Figure 2d. (2) We tested whether TF motifs tended to occur 5’ or 3’ relative to CRX sites and found no significant bias for any TF in either enhancers or silencers.

We added a sentence on p. 11, lines 239-240 stating that “We also did not observe a bias in the linear arrangement of motifs in strong enhancers (Methods).” We describe the analysis in the methods on p. 31, lines 665-671 in the “Predicted Occupancy” section of the methods.

2. p10 The choice of the 8 lineage-defining TFs was somewhat arbitrary because of the arbitrary nature of PWM significance thresholds. Please justify their choice and number, and comment on how well the model performs when this TF set is altered?

The 8 TFs were identified by a de novo enrichment analysis using DREME, as described in the methods on p. 29-30, lines 637-653 in the “Motif Analysis” section of the methods. Aside from these motifs, DREME found no other significant motifs enriched in silencers or enhancers. As described on p. 9, lines 204-206, we also used high-scoring k-mers in the SVM to search for additional motifs that we might have missed in the DREME analysis, but found none.

Sometimes DREME motifs matched PWMs for multiple TFs. We empirically found that replacing a PWM with one that is highly similar (such as MAZ and Sp4, detailed below in 4) does not change model performance, so we chose PWMs that were both high quality and biologically relevant. Following common practice in motif analysis, we excluded low-information content DREME motifs that did not match known TFs, as well as low-specificity dinucleotide motifs not well represented in our sequence classes. While it is always possible that we missed relevant motifs, we performed a thorough search using the standard methods.

We also tested the performance of the model trained on the 8 retina TFs against 100 ‘null hypothesis’ models from randomly selected sets of 8 TFs. In each case the retina TF model outperformed the randomly selected TF model. These results are described on p. 9, lines 199-204 and shown in Figure 2 figure supplement 3.

3) Does an evolutionary change in information content calculated between orthologous *Mus musculus* and, say, Mus spretus sequence help to separate active (enhancer/silencer) sequence from inactive sequence? (https://doi.org/10.1038/sdata.2016.75)

We thank the reviewer for this interesting suggestion. We identified orthologous sequences using an alignment between *M. musculus* and *M. spretus* from the UCSC browser, and generated information content scores for them. However, the 164 bp orthologous sequences have a median of 98% nucleotide similarity, and 75% of orthologs have 94% nucleotide similarity. Because the sequence similarity is so high, there is almost never an evolutionary change in information content between these two species.

p9 Please explain further "we chose to represent the zinc finger motif with MAZ based on the PWM qualities". line 132 "For motifs that matched multiple TFs, we selected one representative TF for downstream analysis (Figure 2—figure supplement 2, Methods)". Please explain further.

We have clarified this point with two changes. On p. 8, lines 175-178 we now state that “we chose to represent the zinc finger motif with MAZ because it has a higher quality score in the HOCOMOCO database (Kulakovskiy et al., 2018).”

On p. 30, lines 649-651 in the methods we explain why we selected one motif per TF family, with the sentence “Because PWMs from TFs of the same family were so similar, we used one TF for each DREME motif, recognizing that these motifs may be bound by other TFs that recognize similar sites.”

Rationale: Sp4 and MAZ are both C2H2 zinc fingers with nearly identical PWMs in the HOCOMOCO database. The Sp4 PWM has a “B” grade quality in the HOCOMOCO database, while the MAZ PWM has an “A” grade, indicating that the MAZ motif is higher quality. Furthermore, the MAZ PWM has a length of 11, while the Sp4 PWM has a length of 16; the extra 5 positions all have very low information content. We therefore chose to represent the C2H2 zinc finger motif with MAZ, though the sites may be bound by Sp4 in photoreceptors.

More generally, TFs that belong to the same family often have PWMs that are highly similar, if not indistinguishable. For example, the de novo motif identified with DREME matching NeuroD1 (Figure 2—figure supplement 2a, bottom of second column) also matches ATOH1, NeuroD2, NGN2, OLIG2, ASCL2, and 40 other PWMs, most of which are E-box binding TFs in the “Tal-related factors” family in the HOCOMOCO database. Although paralogous TFs can have differential binding specificities, these differences are not always captured by PWMs. Additionally, our predicted occupancy metric is a sigmoidal function, so it is insensitive to subtle differences between similar PWMs. Including these subtle differences would have yielded redundant information, so we selected only one motif for each de novo motif identified with DREME.

p21 Why were different FIMO p-value thresholds applied?

This is an error we made during the library design process, though in practice it has no impact on the overall motif content of the selected CRX ChIP-seq peaks versus ATAC-seq peaks. We intended to use a FIMO p-value threshold of 2.5 x 10^-3^ throughout library design, but did not notice the error until after the library was ordered.

We added an additional paragraph on p. 24, lines 497-503 of the “Library Design” section in the methods to explain this error:

“We unintentionally used a FIMO p-value cutoff of 2.3 x 10-3 to identify CRX motifs in CRX ChIP-seq peaks, rather than the slightly less stringent 2.5 x 10-3 cutoff used with ATAC-seq peaks or mutating CRX motifs. Due to this anomaly, there may be sequences centered using ShapeMF that should have been centered on a CRX motif, and these motifs would not have been mutated because CRX motifs were not mutated in sequences centered using ShapeMF. However, any intact CRX motifs would still be captured in the residual information content of the mutant sequence.”

We also reference this new paragraph on p. 22, lines 457-460 by clarifying the language to read “These sequences were scanned for instances of CRX motifs using FIMO version 4.11.2 (Bailey et al., 2009), a p-value cutoff of 2.3 x 10^-3^ (see below), and a CRX PWM derived from an electrophoretic mobility shift assay (J. Lee et al., 2010).”

p27 line 562 "silencers as negatives"?

We have corrected the text accordingly to read “silencers as negatives” rather than “silencers and negatives.”

Reviewer #2:2.1. “I'm not convinced that anything can be determined about silencers using this experimental design.”“I'm concerned that the silencers they detect could be an artifact of the experimental design. The promoter contains CRX sites and NRL sites, so there is some level of basal expression; the silencers are enriched in repressors, so is it just that the elements containing a repressor are silencing the basal transcription?”“I don't think it is possible to make conclusions about a cis-reg element's silencer capacity based on these experiments.”“If the endogenous promoters don't contain CRX or NRL sites or don't have a basal level of transcription, would your element really be a silencer? I don't think it would, and I'm concerned that the results you're seeing are an artifact of your experimental design.”

These comments suggest that reporter assays with an active basal promoter are not a valid method for identifying silencers, and that our results are due to the presence of general repressor motifs in some sequences. Since this critique questions the entire basis of our study, we have an extensive response below, along with a list of our revisions.

1. For both enhancers and silencers, reporter gene assays are a widely accepted method used in high-throughput screens and validation of predicted enhancers and silencers. Reporter gene assays, whether for enhancers or silencers, work by testing the effect of a sequence on the activity of a basal promoter. The reviewer raises an important point that a limitation of these assays is that they are conducted with a general basal promoter, rather than the native promoters on which enhancers and silencers act. In spite of this limitation, reporter assays are a well-established method for discovering and validating both enhancers and silencers. Silencers have been identified using reporter gene assays since they were first discovered over 35 years ago (Brand, et al., *Cell* 41:41-48).

Unlike many reporter assays in the literature, our basal promoter is a natural photoreceptor promoter, which we assayed in living, explanted retina, not in a cell line. Our reporter system is more physiologically natural than many in the literature. We have also observed silencing with other basal promoters in our prior work (see point 2.2 below).

Finally, our reporter gene experiments are in line with recent, high-profile studies of silencers. A moderately high level of basal transcription from a minimal promoter is needed to detect silencing, and all of these studies use an active basal promoter, often a synthetic one:

1. EF-1a promoter driving caspase9 (Pang and Snyder, Nat. Genetics 52:253-263).

2. “Super core promoter” (SCP1) driving GFP (Doni Jayavelu, et al., Nat. Comm. 11:161).

3. Strong enhancer upstream of the HSP70 promoter driving GFP (Gisselbrecht, et al. *Mol Cell* 77:324-37).

4. “HS2” from the human β globin LCR upstream of the SV40 promoter driving luciferase (Petrykowska, et al. Genome Res. 18:1238-46).

5. An enhancer placed upstream of the SV40 promoter driving luciferase (Huang, et al., Genome Res. 29:657-67).

We added two paragraphs of discussion of the literature on silencers in the introduction (page 3-4, 2nd and 3rd paragraphs, lines 42-75, beginning with “Another challenge…”). We cite all of the reporter gene assays above, and note that their findings are similar to ours: silencers are difficult to distinguish from enhancers because they often reside in open chromatin and bear the epigenetic marks of enhancers.

We explicitly reference our prior papers showing silencing is not dependent on the *Rho* promoter, on p. 14, lines 307-310, beginning with “This confirms that the distinction between silencers and enhancers does not depend on the *Rho* promoter…” On the same page, lines 306-307, we also added text noting that less than 3% of autonomous sequences are from the silencer class.

Additionally, we have added a new analysis to support the idea that these silencers function as such in the genome (see below).

2. The presence of general repressor sites does not account for our results – there is extensive evidence that CRX itself is necessary for silencing. The reviewer suggests that silencing may be due to a general effect of repressor motifs acting on the basal promoter, but only 13% of our silencers sequences contain the GFI1 repressor motif (Figure 2c). We found no enrichment of other repressor motifs. Rather than silencing being the effect of repressor motifs, there is strong evidence that CRX itself is required for silencing.

CRX is expressed in photoreceptors and bipolar cells, which encompass more than 15 different cell types with divergent transcriptional programs (Murphy et al., *eLife* 8:e48216). The role of CRX is to selectively activate and repress cell type-specific genes, in cooperation with different cofactors. For example, CRX interacts with other TFs to repress cone photoreceptor genes in rods (Peng et al. *Hum Mol. Genet.* 14:747-64). A massively parallel reporter gene assay in live retina, with a native photoreceptor basal promoter, is a physiologically relevant way to study CRX-dependent silencing at scale.

We previously showed that CRX is necessary for silencing: silencers in our reporter assay were de-repressed in *Crx^-/-^* retina (White, et al. *Cell Rep.* 17:1247-54). In the same study we found that CRX ChIP-seq peaks with sequence features of silencers were near genes that are de-repressed in *Crx^-/-^* retina. We showed then and in the current study that the silencing effect depends on intact CRX sites (Figure 5a).

The specific mechanism of CRX-directed repression is unclear, though there are other examples of transcriptional activators behaving as repressors in the same cell type, acting at binding sites that mediate both activation and repression. GATA-1 is an activator in K562 cells, positively interacting with the co-activator CBP, but GATA-1 also represses the *GATA-2* promoter by displacing CBP and GATA-2 (which auto-activates its own promoter). Sequence context is critical here: only a subset of GATA binding sites mediate GATA-1 based repression, for unknown reasons (Grass, et al. *PNAS* 100:8811-16). Our work bears directly on this question of how sequence context governs whether TF binding sites act to enhance or silence transcription.

To test the genomic relevance of our reporter assay results, we added a new analysis. Similar to our previous work (White, et al. *Cell Rep.* 17:1247-54), we looked at the proximity of our MPRA silencers and enhancers to genes that were up-regulated or down-regulated in *Crx^-/-^* retina. Consistent with the reporter assay results, we found that genes that are *down-regulated* upon loss of CRX are likely to be near sequences we classed as enhancers, while *up-regulated* genes are more likely to be near silencers (odds ratio 2.1). These results support our finding that silencers in the reporter assay act as CRX-dependent silencers in the genome. We describe this result on p. 6-7, lines 133-139, beginning with “To test whether these sequences function as CRX-dependent enhancers and silencers in the genome…” We added details on the analysis to the methods on p. 29, lines 626-635 under the heading **“**RNA-seq Analysis**”**.

We added a more extensive description of our prior work on CRX-dependent repression in the discussion on p. 17-18, lines 387-402, beginning with “Consistent with this, we previously showed…”

3. Repressors in the retina act on CRX motifs and other homeodomain motifs. Rather than repressor motifs, it is CRX motifs in a particular sequence context that are critical for silencing. Other repressors (such as Vsx2) have been shown to bind CRX or RAX homeodomain sites (Dorval, et al. *J Biol Chem.* 281:744-51; Clark, et al. *Brain Res* 1192:99-113), or interact with CRX itself (Sanuki, et al., *FEBS Lett.* 584:753-8; Hlawatsch, et al. *PLOS One* 8:e60633). A combination of CRX and RAX sites in the *Gnb3* promoter selectively silences this promoter in bipolar cells (Murphy et al., *eLife* 8:e48216). Additionally, studies of photoreceptor repressors often use a reporter gene with the *Rho* basal promoter (for example, Sanuki, et al., *FEBS Lett.* 584:753-8; Cheng, et al., *Hum Mol Genet* 13:1563-75; Mitton, et al., *Hum Mol Genet* 12:365-73). Our use of the *Rho* basal reporter to study CRX-dependent activation and repression has deep support in the literature.

Silencers in our assay have high CRX/homeodomain motif content, which could make them targets for other photoreceptor-specific repressors. In *Drosophila* photoreceptors, selective silencing of opsin genes is achieved by a repressor, Dve, which binds the motifs of Otd, an ortholog of CRX (Rister, et al. *Science* 350:1258-61). A similar mechanism may operate at some genes in the mammalian retina.

We added a discussion of the potential mechanisms of repression by CRX, and of the evidence that other TFs act through CRX motifs, on p. 18, lines 404-421, the paragraph beginning with “The contrast in motif diversity…”.

2.2. “Moreover, what does this mean relative to the elements in the endogenous locus, if an endogenous promoter doesn't have CRX or NRL sites within its promoter or basal transcription does this mean the silencers as described in this assay are not really silencers within the genome.”“As silencers are decreasing expression from a basal promoter and the whole paper is centered on this, I think the choice of promoter in this experiment is critical. I don't think one promoter can be used to draw such conclusions… It would be helpful to test these elements with the Rho promoter where you mutate the NRL and CRX sites. It would also be helpful to test these elements with a different promoter for another gene that is expressed in the retina, ideally one that doesn't contain CRX and NRL sites, to see if there are enhancer-promoter interactions that are influencing your results and thus your conclusions.”

As noted in 2.1, we now report supporting evidence that CRX-dependent silencers act as such in their genomic context: silencers are enriched near genes that are de-repressed in *Crx^-/-^* retina.

Additionally, we have two responses to the reasonable suggestion to try other promoters:

1. We tested other promoters in our prior work and observed silencing. In White, et al. *Cell Rep.* 17:1247-54, Figure 2B, we showed an NRL motif is not necessary for silencing. Sequences with high CRX predicted occupancy were inactive with a promoter lacking an NRL motif. In Hughes, et al. *Genome Res.* 28:1520-31, Figure 2B and Supplemental Figure 8, two of us (J.C.C. and C.A.M.) showed that silencing occurs with the *Crx* promoter.

2. CRX motifs are very highly enriched in photoreceptor promoters. Testing our silencer with a promoter lacking CRX sites is not relevant, since we make no claim that these are general silencers. Our aim is to understand how CRX selectively regulates its targets. As we discussed above in 2.1, silencing depends on CRX, or possibly on repressors that bind CRX sites. CRX motifs are very highly enriched in open chromatin in photoreceptors and bipolar cells (Murphy et al., *eLife* 8:e48216) and common in photoreceptor promoters (Hsiau, et al., *PLOS One* 2:e643).

We explicitly cite our prior work showing silencing with promoters other than *Rho* on p. 14, lines 307-310, the sentence: “This confirms that the distinction between silencers and enhancers does not depend on the *Rho* promoter, and the result is consistent with our previous finding that CRX-targeted silencers repress other promoters (Hughes et al., 2018; White et al., 2016).”

As described in 2.1, we have now included more discussion of prior work and the evidence for the role of CRX to both enhance and silence target genes. This new material is in the introduction (p. 3-4) and the discussion (p. 17-18). In 2.1 above we describe the new analysis added to the paper (p. 6-7) finding that silencers are more likely to be near de-repressed genes in *Crx^-/-^* retina, which supports the claim that these are CRX dependent silencers.

2.3. “I am also not convinced that silencers and enhancers are different things. If a genomic element controls the time and location of gene expression, then it is an enhancer – enhancers can bind activators and repressors and restrict expression to only particular cell types. I think trying to call things enhancers, and silencers makes things overly complex, especially considering the fact the authors point out that the same element can be an enhancer in one tissue type and a silencer in another.”

We use the standard silencer/enhancer terminology in the field. We agree with the reviewer that silencers and enhancers are not necessarily different things. We suggest that the silencers we identify may be enhancers in other retina cell types (p. 18-19, lines 423-429), and we cite multiple recent papers that show this (Pang and Snyder, *Nat. Genetics* 52:253-263; Doni Jayavelu, et al., *Nat. Comm.* 11:16; Ngan, et al., *Nat. Genetics* 52:264-72; Halfon, *Trends Genet.* 36:149-51; Gisselbrecht, et al., *Mol. Cell* 77:324-337). The idea that enhancers also act as silencers has existed in the literature since silencers were first discovered (Brand, et al., *Cell* 51:709-719; Jiang, et al., *EMBO J* 12:3201-3209).

Our use of ‘silencer’ and ‘enhancer’ is consistent with these studies. A 2021 review of silencers (Segert, et al., *Trends Genet.* 6:514-27) describes them as follows:

“Silencers are regulatory DNA elements that reduce transcription from their target promoters; they are the repressive counterparts of enhancers. Although discovered decades ago, and despite evidence of their importance in development and disease, silencers have been much less studied than enhancers. Recently, however, a series of papers have reported systematic studies of silencers in various model systems. Silencers are often bifunctional regulatory elements that can also act as enhancers, depending on cellular context…”

2.4. “Would they find that enhancers as they define them bind a combination of transcriptional activators and that silencers bind some activators such as CRX in combination with a transcriptional repressor expressed in the cell type where the element acts as a silencer?”“The paper claims that information content can be used to distinguish between these two classes of element, I would like to see how this compares to prevalence to transcriptional repressors, and activators found in silencers and enhancers as the only TF mentioned to work with CRX in the silencer is a well-known repressor Snail. Would the presence of a repressive TF be a better predictor of silencer vs. enhancer activity?I would like to see that the information content model performs better than measuring the prevalence of activator and repressor motifs.In line with this, the difference in information content between a silencer and an enhancer is 3 motifs for 2 tfs vs. 3 motifs for 3 tfs or 4 motifs for 2 tfs. For the silencers and enhancers, what % of these TF motifs are repressors, and what % are activators.”

Repressor motifs are not critical for silencers: We searched extensively for enriched motifs but found only one repressor motif, GFI1 (noted on p. 8, line 178-181, with details on p. 29-30 under “Motif Analysis”). This motif is present in only 13% of silencers (shown in Figure 2c). The presence of this single repressor motif is not enough to classify silencers.

We compared the performance of our information content model (AUROC 0.634, Figure 3b) to the performance of a model based on the 8 enriched TF motifs (AUROC 0.698, Figure 2a). Both models perform somewhat worse than the SVM (AUROC 0.781, Figure 2a). In the manuscript on p. 12-13, lines 279-284 we describe why the information content model is significant, despite its lower performance than the two other models:

“[The information content model] is only slightly worse than the model trained on eight TF occupancies despite an eight-fold reduction in the number of features, which is itself comparable to the SVM with 2,080 features. The difference between the two logistic regression models suggests that the specific identities of TF motifs make some contribution to the eight TF model, but that most of the signal captured by the SVM can be described with a single metric that does not assign weights to specific motifs.”

On p. 16, lines 366-371 we provide further interpretation:

“These differences [between enhancers and silencers] are captured by our measure of information content. Information content, as a single metric, identifies strong enhancers nearly as well as an unbiased set of 2,080 non-redundant 6-mers used for an SVM, indicating that a simple measure of motif number and diversity can capture the key sequence features that distinguish enhancers from other sequences that lie in open chromatin.”

TFs are often bifunctional and motifs can’t easily be classified into repressors and activators: As we discuss above in 2.1, silencing depends on CRX protein and CRX motifs (which may be bound by other repressors). It is well known that many TFs, like CRX, function to both activate and repress. It is not effective to simply classify sequences based on an inventory of supposed activator or repressor motifs.

Sequence context matters, which is one of the central claims of our paper. This is why our information content model is significant: it suggests that silencers, at least in photoreceptors, are not defined by repressor motifs, but rather by a lack of diverse binding sites for expressed TFs, which provides the sequence context for CRX-dependent silencing.

As mentioned above, we added more discussion of the evidence for the role of CRX in both repression and activation. This is found in the introduction (p. 3-4, lines 64-75, beginning with “The TF cone-rod homeobox (CRX) controls selective gene expression…”), and in the discussion (p. 17-18, lines 387-421).

We added results for RAX to the motif co-occurrence matrix (Figure 2d) so that the figure now represents all enriched TF motifs.